# Uncertainties in land use data may have substantial effects on environmental planning recommendations: A plea for careful consideration

Felix Neuendorf[1]*, Julia Thiele[1], Christian Albert[2], Christina von Haaren[1]

1 Institute of Environmental Planning, Leibniz University Hannover, Hannover, Lower Saxony, Germany,
2 Institute of Geography, Ruhr University Bochum, Bochum, North Rhine-Westphalia, Germany

* neuendorf@umwelt.uni-hannover.de

**Data Availability Statement:** Data for case study 1 and 2 is published on the Zenodo Data repository (DOI:10.5281/zenodo.5602765) and can be accessed via the URL:

## Abstract

A key challenge of environmental planning is to craft recommendations for future sustainable spatial development amid ubiquitous uncertainties. This paper aims to explore how different data uncertainties, usually unknown to the planner, may influence environmental planning recommendations. We apply a case study-based approach, in which we provide three illustrative examples of how data with different kinds and levels of uncertainty affect environmental assessments and, by that, the decision-support provided by environmental planning. The cases stem from different spatial levels in Germany and consider 'Regional soil-based climate change mitigation' in the region of Hannover, 'State-wide habitat conservation siting' in the federal state of Saxony-Anhalt, and 'National renewable energy planning'. Based on the three examples, we discuss implications for planning practice and derive recommendations for further research. The three cases studies illustrate the substantial effects of data uncertainty on environmental assessments and planning recommendations derived from those results. We identify four problem constellations of dealing with data uncertainty in environmental planning that relate to the severeness of uncertainty impacts, the responsibility of the decision-maker, and the kinds of impacts that wrong decisions may have. We close with recommendations for further research, among others to develop robust and pragmatic methods for identifying the uncertainty levels in environmental data and assessment results.

## Introduction

A key challenge of environmental planning is to craft recommendations for future sustainable spatial development amid ubiquitous uncertainties due to incomplete understandings of the ecological, social and economic conditions, trends, and interrelations [1, 2]. A recent review revealed that the main planning relevant categories of uncertainties in environmental planning can be associated with four categories [3]: Data uncertainties, which are most commonly

https://zenodo.org/record/5602766. Data for Case study 3 (DOI: 10.25835/0072778) is published and available via the URL: https://data.uni-hannover.de/dataset/dataset-areas-with-low-and-medium-spatial-vulnerability-to-a-prototype-wind-turbine.

**Funding:** This study was funded by the German Research Foundation (HA 3301/8-1, 250763334), https://www.dfg.de/en/index.jsp - FN, and the German Federal Agency for Nature Conservation (FKZ 3515 82 4300), https://www.bfn.de/en.html, in cooperation with the German Ministry for the Environment, Nature Conservation and Nuclear Safety, - JT. The publication of this article was funded by the Open Access Fund of the Leibniz Universität Hannover. The funders had no role in study design, data collection and analysis, decision to publish, or preparation of the manuscript.

**Competing interests:** The authors have declared that no competing interests exist.

associated with geometric uncertainties that arise mainly from resolution, thematic uncertainties that occur when values are attributed to objects, and temporal mismatches between data sources [4]. Model uncertainties, the second category, relate to challenges in choosing appropriate indicators and variables for environmental assessments [5]. Models applying only a small number of explanatory variables are generally easier to use for planners, but are often inferior to more complex and dynamic models in understanding environmental system change. The most obvious trade-off in model development for planning purposes is between precision and usefulness, for example when a complex model would become more demanding in data requirements [6]. In addition, a very high level of detail can be impractical and a source of uncertainty in planning applications when the interpretation of complex model results puts high demands on user [7]. The third category, projection uncertainties, is linked to projecting future states of the environment—atypical challenge for planning as a strong forward looking action [8]. Plans often suggest that problems are of a static nature, but in reality they are highly dynamic [9] and can often not be predicted with the understanding of future behaviour of various components [10]. Finally, valuation uncertainties are introduced when norms and laws need to be interpreted to form scales for valuation [3]. While data, model and valuation uncertainties are prominent at multiple stages of the planning cycle, projection uncertainties come into play in particular when scenarios for future developments are created and potential scenario impacts are being explored.

Uncertainty of spatial and statistical data presents one of the most important uncertainties, as such input data inaccuracies propagates through the entire environmental assessment and planning process. Relevant concerns of environmental planning in terms of data uncertainties include, among others, mistakes in categorising (for example, if particular land parcels are incorrectly attributed to land use types), inconsistencies between the different datasets (mainly the location and spatial delineation of features) and impacts of data resolution (for instance if the resolution it too coarse to divide site-specific management recommendations to farmers). Some of these uncertainties may be obvious, but not quantified, others may be hidden in the data set. Both types of uncertainties are usually not explicitly communicated with the data provision. Nevertheless, the planner has to decide which data detail will suffice for which specific planning task—risking misjudgement. As planners are striving for efficiency in the planning process, they tend to use the data set that is least detailed while still fulfilling addressing the planning question with sufficient accuracy and robustness. Also, not uncommon, data has to be used which is obviously not optimal but available, and often planning recommendations necessarily reflect an unquantified or unquantifiable uncertainty.

Over the last decades, environmental planners devised some strategies for providing decision-support amid such unknown or suspected data uncertainties. One is the participation strategy according to which the involvement of diverse knowledge holders in the environmental assessment and planning process can integrate complementary insights and minimize potential bias. However, this strategy seems to be effective only at local level. Most common in theory but costly and thus rarely used in practice is the adaptive management strategy [11, 12] that proposes to frame interventions as hypotheses, to monitor impacts of implementation, and to continuously revise decisions in face of enhanced understandings [13]. In Europe, applying such strategies is also guided by the legally enshrined precautionary principle according to which environmental planners should use caution and additional safeguards in situations where scientific understanding and data certainty is lacking and decisions could potentially cause serious harm to people and nature [14]. Finally, environmental planners often presume that an intuitively felt uncertainty may be present in their basic data, but that this will not relevantly affect the response measures. The logic behind this is that planners often strive for multifunctional measures which intentionally target a larger area than

monofunctional measures that usually address only smaller areas, for example to safeguard a certain species. Along this line, planners generally presume that the inaccuracies of land use data will not affect the validity of recommended measures which address multiple purposes in an aggregated spatial unit. Furthermore, in case of precautionary measures, the uncertainties of proposals of future land uses seem to be much more relevant than uncertainties stemming from inaccuracies of cartographic land use data.

The problem remains, however, that environmental planners usually miss comprehensive understanding of the kind and degree of severity of the uncertainty embedded in their data, and how this might affect their assessment results which constitute the basis for their recommendations. In consequence, planners remain unsure if the above outlined uncertainty tackling strategies are needed, if they work or if opportunities for better targeted or efficient recommendations are missed. We hypothesize that a lack of uncertainty information or awareness may lead to substantial mismatches between actions proposed and needed, and that providing more uncertainty information about geodata would allow for better-adapted planning recommendations.

This paper aims to explore how different data uncertainties, usually unknown to the planner, may influence environmental planning recommendations. We apply a case study-based approach, in which we provide three illustrative examples of how data with different kinds and levels of uncertainty affect environmental assessments and, by that, the decision-support provided by environmental planning. Based on the three examples, we discuss implications for planning practice and derive recommendations for further research.

## Methodological approach

Our research method follows a typical qualitative and comparative case study analysis [15]. In order to test our hypothesis, we selected three case studies from different levels of environmental planning decision making in Germany (Table 1). The first case study, 'Regional soil-based climate change mitigation', concerns environmental planning for Greenhouse Gas (GHG) emission reduction from soils in the region of Hannover. The case study focuses on identifying

**Table 1. Data applied in the case studies.**

| Case study | Case study location / model run | Land use / Land cover classification | Date of mapping / release date | Reference size / spatial accuracy | Number of different LC classes |
|---|---|---|---|---|---|
| First case study: **'Regional soil-based climate change mitigation'** | Location: Lower Saxony | Biotope-mapping | 2007–2011 | - | 557* |
| | | ATKIS (ni-LBM) | 2009 | 1ha | 44 |
| | | CORINELand Cover | 2012 | 25ha | 30 |
| Second case study: **'State-wide habitat conservation siting'** | Location: Saxony-Anhalt | Biotope-mapping | starting 2009 | - | 184* |
| | | ATKIS (Basis-DLM) | Starting2011** | *3–5 m* | 50 |
| Third case study: **'National renewable energy planning'** | First model run | • DLM250 | 2015 | 100 m | 36 |
| | | • DEM200 | 2014 | Grid cell size 200 m; Accuracy: Planimetry:± 1–3 m; Altimetry: ± 3–10 m | - |
| | Second model run | • ATKIS (Basis-DLM) | 2018 | *3–5 m* | 50 |
| | | • DEM50 | 2017 | Grid cell size 50 m; Accuracy: Planimetry: ± 3 m; Altimetry: ± 2–4 m | - |

* Federal states use different classification manuals which result in different number of classes on a comparable level.

** Very high actuality on classes that are listed as "up-to date", mainly infrastructure or demarcations with legal consequences.

priority areas for climate protection and management. Potential users of such information would be the state or the regional and local governments. The second case study, 'State-wide habitat conservation siting', addresses the identification of valuable habitats for protection planning in the German federal state of Saxony-Anhalt. Case study number three, 'National renewable energy planning', concerns an environmental planning effort aiming to identify areas for the siting of renewable energies, which should also guide decision-making at the regional level.

For each case study, we apply two or three data sets and compare the assessment results in which the data is used, and the planning recommendations deduced from the assessment results. We compare resulting maps also in terms of spatial consistency, meaning areas where there is a spatial match of class labels between different outputs. One of the data sets applied in each case represent data easily available at the respective planning level that might thus be chosen by planners as the basis for assessments despite of having no detailed information about inherent uncertainties—which is a realistic assumption (easy data variant). The second or third data set applied represented the best available data and was used as a reference (reference variant). The juxtaposition of the planning outputs based on data of different levels of uncertainty, and an exploration whether the uncertainty of the less detailed data would be acceptable for the planning purposes, will provide us input for the reflection on implications of data uncertainties on environmental planning recommendations in the discussion section. In addition, we explored whether knowledge about structural uncertainties or errors in the dataset could lead to better adapted planning recommendations even if the easily available data set is used.

## Data sources and properties in the three case studies

Our first two case studies, 'Regional soil-based climate change mitigation' and 'State-wide habitat conservation siting', draw on two or three kinds of spatial datasets commonly used in different sectors of spatial planning in Germany (see Table 1). As the first and coarsest dataset, we used CORINE Land Cover 25 (CLC25) data. It represents the European standard for land cover classifications and was introduced in 1985 as part of the Copernicus programme. It is being updated every six years using the latest remote sensing technologies. We used the 2012 dataset that has been generated using IRS P6 LISS and rapid eye data and has a minimum mapping unit of 25 hectares. The CLC 25 data of 2012 has a proclaimed thematic accuracy of > = 85% with the addition of the phrase "probably achieved" [16].

The cadastral data, called ATKIS (Digital landscape model—base data), is the second data source used in our case studies. ATKIS data currency differs between federal states with a maximum time span of five years between complete updates. In Saxony-Anhalt, the base data set is updated every three years to ensure a higher degree of topicality. Selected land use/land cover classes such as roads, water bodies and protected areas are updated every three to twelve months. Thus, in most cases, the ATKIS dataset represents the most up-to-date information on land use and land cover. The specified positional accuracy is three to five meters. For Lower Saxony, we used the data of Lower Saxony land cover model, which resembles the ATKIS base data but has less thematic classes and a minimum mapping unit of one hectare.

Biotope mapping data, the third dataset, is used as a reference, as it is the most detailed type of land-use data used for environmental planning in Germany and we assume that it depicts the real circumstances best. Biotope mappings are generated using a combination of remote sensing, mainly interpretation of aerial images, ATKIS data and field surveys. There is no minimum mapping unit for biotope mappings, other than that the biotopes should be easily distinguishable in the field [17]. However, also this basic data set cannot be considered completely

reliable, as it is not updated as often as ATKIS data. Furthermore, while spatial accuracy of the biotope delimitation can be considered relatively similar across federal states, the assessments differ in their class hierarchies and in specific thematic information given within each class [17, 18]. This highlights the importance of the uncertainty analysis of different data sources, and of describing the sources of uncertainty to be expected in each data set.

For the purpose of the assessment of inconsistencies between the data sets, the classes have been aggregated. We opted to aggregate classes with high relevance for landscape planning into groups that best reflect important areas, the baseline being higher aggregation levels in the biotope mapping. A different aggregation scheme might be used when addressing inconsistencies in another branch of planning. In difficult cases, experts were consulted to ensure accurate classifications.

## Assessment methods applied in the case studies

The case study 'Regional soil-based climate change mitigation' illustrates the impacts of applying two sets of commonly used land use data with different levels of consistency on recommendations for minimizing GHG emissions from soils in the Region of Hannover. The land use data sources used included ATKIS data, and CLC 25 data and for reference, biotope mapping data. For each dataset, we applied the same method for estimating potential $CO_2$-emission and retention potentials [19]. The applied method combines data about soil types and land-use/land-cover which are classified as to climate relevance, and should be available for a coherent time span of 5 years. The investigation was conducted for the reference year 2015, i.e., the results show the potential $CO_2$ retention function for this year. For this purpose, the biotope type mapping was extended with information from the LEA portal (SLA) on the agricultural impacts of the years 2010–2015. For the investigation of the ATKIS data, it had to be assumed that both grassland and arable land have existed for more than 5 years due to a non-optimal data situation.

The case study example 'State-wide habitat conservation siting' focussed on the impacts of using data sources of different uncertainty levels when identifying valuable habitats for species protection planning in the German federal state of Saxony-Anhalt. We used ATKIS data, and biotope mapping data as reference. We identified habitats of very high conservation value in the biotope mapping dataset, and compared the consistency with respective classes in the ATKIS data. The considered land use classes include 'fen, bog and marshlands', 'heathlands', and 'small natural standing waters'. We expected a higher inconsistency in the latter class, because ATKIS data only has standing water as a delimitable class and no distinctive features regarding the area size. We applied descriptive statistics to explore commonalities and differences, an analysis method that can be easily communicated and comprehended by different addressees.

The case study 'National renewable energy planning' explored impacts of using datasets of different uncertainty levels on the potentially suitable sites for renewable energies, taking into account both human and nature-conservation needs [20]. We used the data and results of two model runs, the first one was conducted by Walter et al. 2018 and the second one by Thiele et al. 2021, for an uncertainty analysis in order to quantify the influence of data resolution on identifying opportunity spaces for generating renewable energy.

In the first run of modelling the area potentials for renewable energy, data was used which are easily available on national scale [21], namely the digital landscape model 1:250,000 (DLM250) and the digital elevation model 200 (DEM200). These spatial data were considered sufficient for generating a satisfactorily accurate estimation of opportunity spaces for renewable energy generation that are compatible with human and nature conservation

considerations. It was also assumed that the data accuracy was sufficient to scale down the spatial results to the level of federal states. The DLM250 was used to select, for example, settlements, infrastructures, water bodies or arable land. Areas with slopes greater than or equal to 30˚ were determined as not suitable sites for wind turbines [21]. These areas were identified in the first run with the digital elevation model (DEM200), which has a grid cell size of 200 m. The height accuracy of this low-resolution model is +/- 20 m.

In order to define the inaccuracies of the results due to the data resolution, a second model run with more precise data was conducted [20, 22], namely ATKIS Basis-DLM and the DEM50 digital elevation data. For the selection of land use types such as settlements and water bodies, the ATKIS Basis-DLM was applied, which provides a much higher surface sharpness with a resolution of 1:25,000 than the lower-resolution DLM250 used in the first run. The second run also applied the DEM50 elevation data with grid cell sizes of 50 m and a height accuracy of +/- 2.5 m to 10 m, instead of the coarser one from the first run. We compared the spatial results by quantifying differences in the spatial input data, and by subtracting the result dataset of the first model run from the result dataset of the second model run. In order to compare the two digital elevation models (input data), the DEM200 was converted to a grid with a grid cell sizes of 50 m. Since the potential areas for onshore wind energy of the two model runs were also provided as grids with a grid cell size of 50 m, they could be compared without further transformations.

## Results

The results of the assessment performed in the case study 'Regional soil-based climate change mitigation' (Fig 1, Table 2) show dramatic differences between the modelled quantities of $CO_2$ depending on the data sets used [23]. Differences in the spatial delineation of peatland and bog soil areas between ATKIS or CLC 25 data are most important here, as the quantitative $CO_2$ emission risk and potential of those soil types are much higher than that of other hydromorphic soils (>1700 versus >70t $CO_2$). While the overall consistency of ATKIS and CLC 25 data is relatively high with 76% and 80%, respectively, substantial differences in the $CO_2$ assessment results emerge depending on which data source and associated uncertainties are applied (see Table 2). The large discrepancies between the datasets highlight the importance of a good representation of sites of importance for climate protection. The consistency of areas of highest level of climate protection (level 5) is substantially lower, with ATKIS data only achieving 31% and CLC 25 data achieving 44% of consistency compared with the reference data. The decline in correctly represented area of around 2,267 hectares when using ATKIS data corresponds to about 4.2 million t $CO_2$ retention potential at risk to not be considered in planning proposals. Surprisingly, using coarser CLC 25 data leads to better results, with a smaller area deficit of around 1,472 hectares that accounts for around 2.8 million t $CO_2$ retention potential untapped.

The use of ATKIS data results in an underestimation of about 5.8 million tons of potential $CO_2$ emissions compared to the reference data. This is because a total of 3,025 fewer hectare were mapped as Level 1 which corresponds to around 44% of the features labelled consistently in comparison to the reference data. In contrast to the highest retention level, the consistency levels further decrease, for example for CLC 25 data set with only 18%. This results in a loss of around 5,800 hectares in contrast to the 7,840 hectares in climate protection level 1 at the reference scale, which accounts for a potential underestimation of the $CO_2$ emission potentials of around 9.7 million tons.

These observations are very important when looking at the overall climate balance for the region. The use of our reference dataset suggests a climate balance would trend more towards

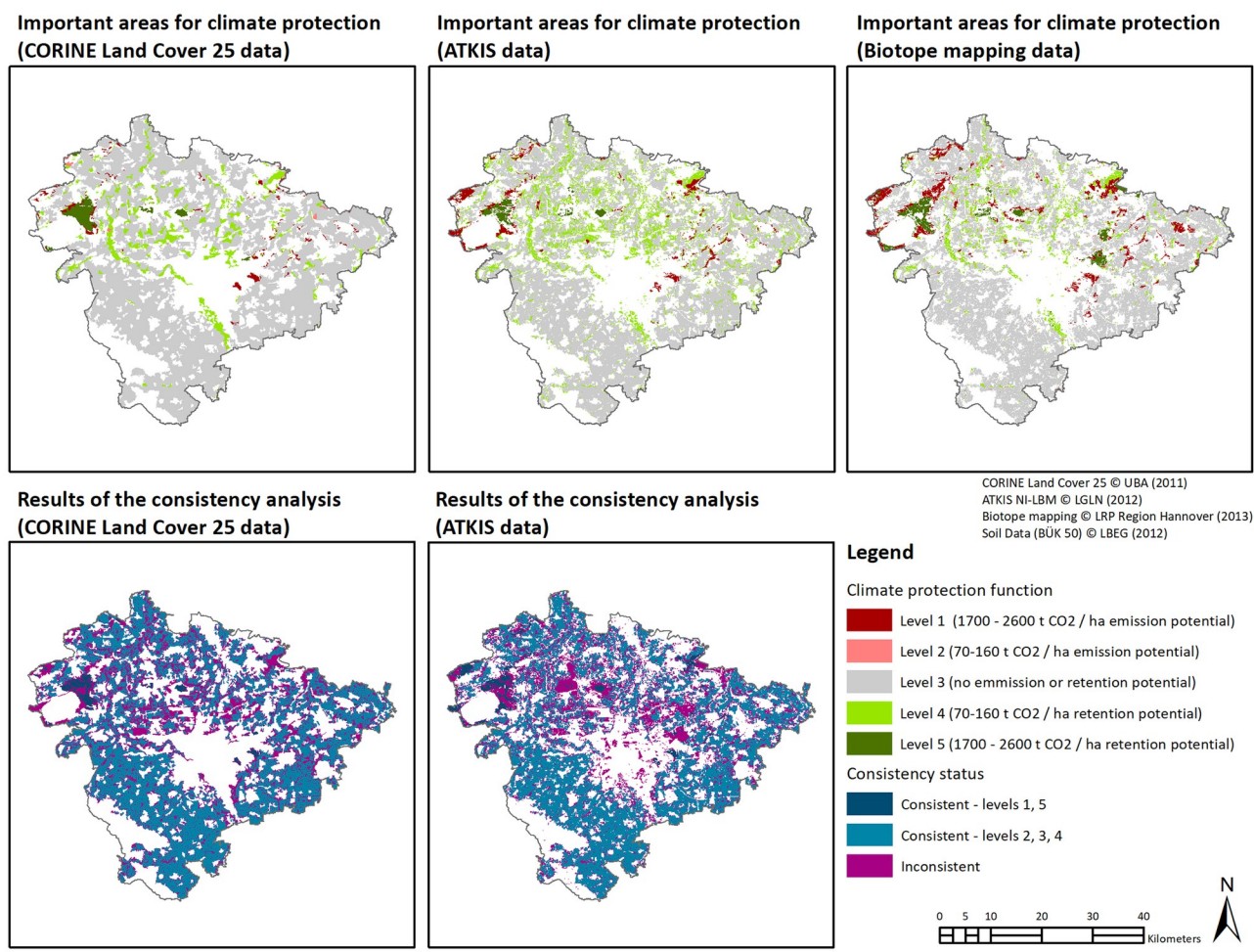

**Fig 1. Spatial representation of the results of the case study 'regional soil-based climate change mitigation'.**

a highlighted potential emission of around 6.5 million t $CO_2$. When using ATKIS data this would be reduce by about 50% to an emission potential of about 3.3 million t $CO_2$. Using CLC 25 data the statement would switch completely, in that the planners would assume a retention potential of about 830,000 t CO2.

Results of the case study on 'State-wide habitat conservation siting' (Fig 2, Table 3) illustrate that about 413 hectares of area with a need for conservation protection according to Biotope mapping data are not correctly represented when using ATKIS data (II + IV). However, about 58 hectares of those incorrectly mapped areas are at least attributed to another type of area with conservation protection needs. This would lead to a total of 355 ha of areas important to conservation not recognized in planning recommendations due to the lower detail of the data used. As expected, a large share of about 3,570 hectares of areas are wrongly identified as areas that require protection status, mainly due to larger standing water bodies represented in the ATKIS data.

Results for the case study 'National renewable energy planning' show that the area potentials for generating renewable energy are reduced when applying higher-resolution data: According to the first model run, 8,300 km$^2$ (2.3% of the area of Germany) are usable for onshore wind energy in a way that is compatible with humans and nature. The second model

**Table 2. Quantitative results of the case study 'regional soil-based climate change mitigation'.**

| Climate protection function according to Saathoff et al. 2013 | | Level 1: Potential $CO_2$ emission of 1700–2600 t/ha | Level 2: Potential $CO_2$ emission of 70–160 t/ha: | Level 3: No emission/ retention | Level 4: Potential $CO_2$ retention of 70–160 t/ha | Level 5: Potential $CO_2$ retention of 1700–2600 t/ha | Overall (ha) / $CO_2$ Balance (t) [- = Retention. + = Emission] |
|---|---|---|---|---|---|---|---|
| | Data set | LU/LC: Arable land and Grassland Soiltypes: Peat and bog soils | LU/LC: Arable land (<5 years old) Soil types: all other hydromorphic soils | LU/LC: Arable land (>5 years old) and Grassland (<5 years old) Soil types: all other soils | LU/LC: Grassland (>5 years old) Soil types: all other hydromorphic soils | LU/LC: Peatland and bogs Soil types: Peat and bog soils | |
| **Biotope Mapping** *Reference dataset* | Surface area in ha | 7,842 | 733 | 96,154 | 10,518 | 4,155 | **119,401** |
| | Potential CO2 emission or retention in t | 14,798,341 | 67,905 | 0 | 986,309 | 7,418,954 | **6,460,982** |
| **ATKIS** | Surface area in ha | 4,817 | 0 | 98,993 | 21,897 | 1,887 | **127,594** |
| | Difference to reference in ha | -3,025 | -733 | +2,839 | +11,380 | -2,267 | **+8,193** |
| | Consistent area in ha (same lvl as in reference) | 3,455 | - | 83,344 | 6,470 | 1,279 | **91,093** |
| | Consistency in % | 44.0 | 0 | 86.7 | 61.5 | 30.8 | **76.3** |
| | Potential CO2 emission or retention in t | 8,941,704 | 0 | 0 | 2,384,509 | 3,261,472 | **3,295,723** |
| | Deviation from reference regarding the emission/ retention potential in t | -5,856,637 | -67,905 | 0 | +1,398,199 | -4,157,482 | **-3,165,259** |
| **CLC 25** | Surface area in ha | 2,262 | 370 | 118,127 | 12,487 | 2,683 | **135,928** |
| | Deviation from reference regarding the in ha | -5,580 | -363 | +21,973 | +1,969 | -1,472 | **+16,527** |
| | Consistent area in ha (same lvl as in reference) | 1,403 | 3 | 88,735 | 5,043 | 1,830 | **95,608** |
| | Consistency in % | 17.9 | 0.4 | 92.3 | 47.9 | 44.0 | **80.1** |
| | Potential CO2 emission or retention in t | 5,070,617 | 66,242 | 0 | 1,387,255 | 4,579,275 | **-829,671** |
| | Deviation from reference regarding the emission/ retention potential in t | -9,727,724 | -1,663 | 0 | +400,946 | -2,839,680 | **-7,290,653** |

run shows a human- and nature-friendly usable area potential of 5,320 km$^2$ (1.5% of the area of Germany) [24]- a reduction of 36%, 2,980 km$^2$ [22] (Fig 3).

Compared to the first model run, industrial and commercial areas, for example, increase by 241% (4,616 km$^2$) in the second model run. Recreation and leisure areas even increase by 3,561%. This can be explained by the fact that smaller industrial or recreational areas are represented in the ATKIS Basis-DLM, but not in the DLM250. The selected built-up areas of the Basis-DLM are slightly smaller (6%, 1,687 km$^2$) than in the selection from the DLM250, but the selection of the Basis-DLM is considerably more exact and detailed. For this reason, after a 750-m-bufferfor noise protection was added to the built-up areas, the area in the second

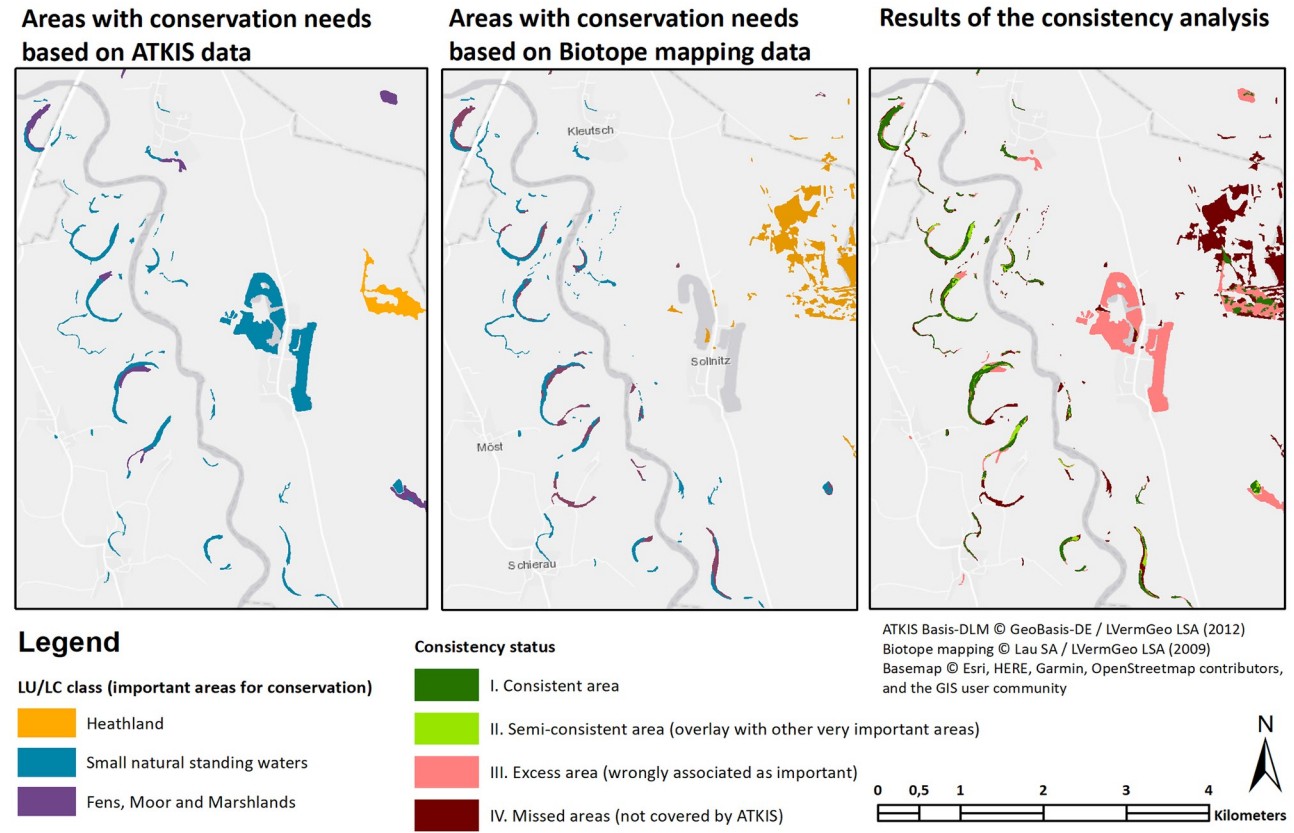

**Fig 2. Spatial representation of selected results of the case study 'state-wide habitat conservation siting'.**

model run (Basis-DLM) increases by 123% (151,414 km$^2$) compared to the first model run (DLM250) (Fig 4).

Infrastructures (e.g. roads, railroad lines), which are available as line features in the DLM250 as well as in the ATKIS Basis-DLM, could also be represented in a more detailed manner in the second model run due to the fact that the ATKIS Basis-DLM offers a variety of attributes compared to the DLM250. Road axes could thus be buffered according to width

**Table 3. Quantitative results of the case study 'state-wide habitat conservation siting'.**

| | Type of area considered | Fens, moor and marshlands (ha) | Heathland (ha) | Small natural standing waters (ha) | Total area (ha) |
|---|---|---|---|---|---|
| **Biotope mapping (very important areas)** | Reference data (most accurate data on biotopes) | 247.45 | 388.72 | 103.48 | **739.65** |
| **ATKIS Basis-DLM** | Comparison data | 313.72 | 633.84 | 3003.84 | **3981.40** |
| **Assessment of inconsistencies** | I. Areas with consistent labels | 74.90 | 179.87 | 71.58 | **326.35** |
| | II. Areas with semi consistent labels (matching other very important areas) | 5.64 | | 52.10 | **57.74** |
| | III. Excess areas (areas that are wrongly associated as important in ATKIS) | 233.18 | 453.97 | 2880.15 | **3567.30** |
| | IV. Missed areas (areas identified as important areas in the reference data, but not identified as such in ATKIS data) | 120.82 | 204.62 | 30.12 | **355.56** |

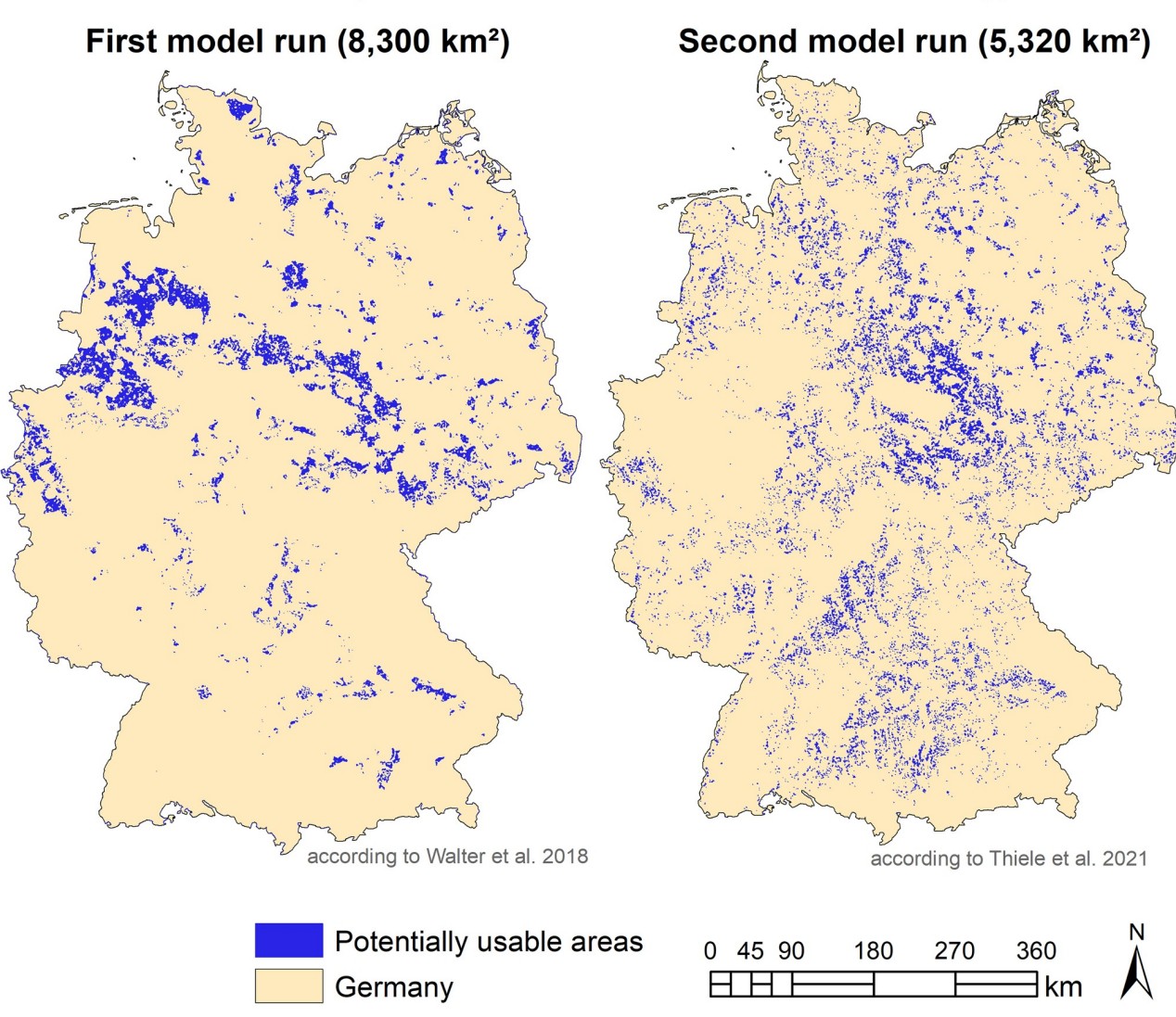

**Fig 3. Comparison of the potentially usable areas for onshore wind energy (left: First model run; right: Second model run [24]).**

classes and railroad lines were buffered according to their number of tracks (2.5 m single-track railroad line, 4.5 m double-track railroad line) [20, 22].

Areas with slopes greater than or equal to 30° record an increase of 1,150%(6,570 km$^2$) in the second model run (DEM50) compared to the first model run (DEM200). The difference between the selections of the two digital elevation models with identical SQL expression is particularly visible when zooming in on low mountain ranges: according to the DEM200, for example, almost no areas with a slope of $\geq$ 30° are found in the Harz Mountains [22] (Fig 5).

We can summarize for the case 'National renewable energy planning', that we observed the same effect on planning-relevant information as in the case of 'Regional soil-based climate change mitigation': The seemingly small error for instance in the spatial data for built up area for Germany has a much bigger effect when included in an assessment. The area excluded for wind turbines increases, as also a small hamlet (not considered in theDLM250 data set) has to be surrounded by a 750m buffer zone for noise protection.

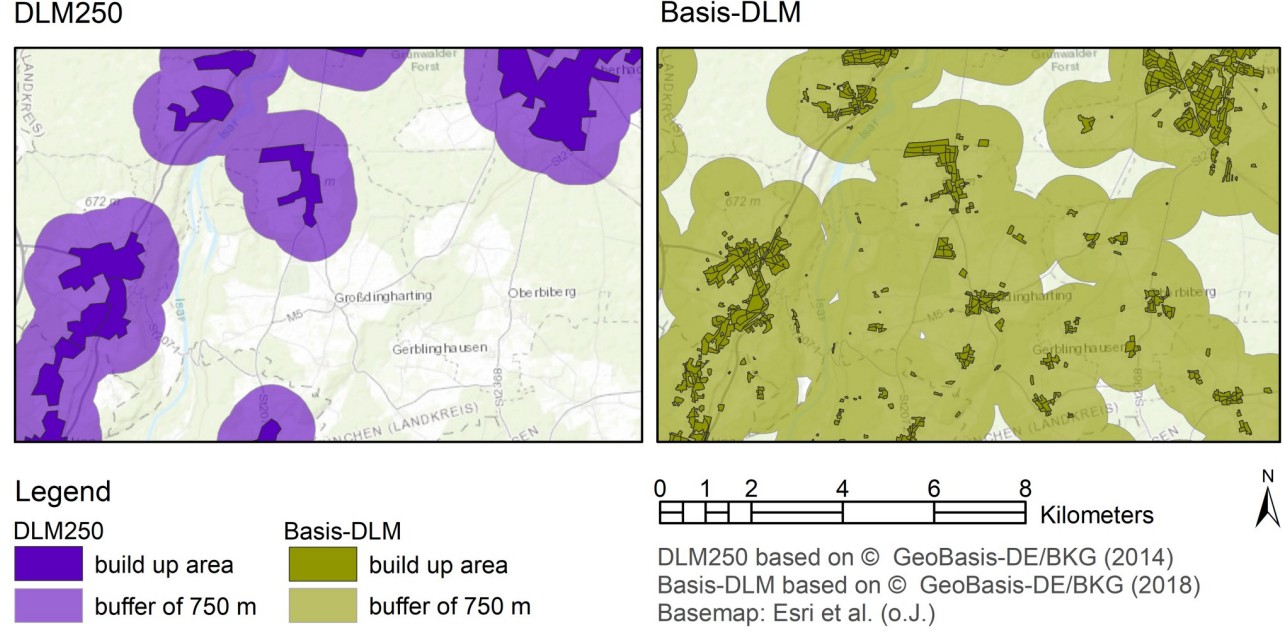

**Fig 4. Comparison of the selection of built-up areas from the Basis-DLM (left) and DLM250 (right) with a noise buffer of 750 m exemplified by the municipality of Straßlach-Dingharting south of Munich.**

## Discussion and conclusion

This paper has provided three case studies from landscape and environmental planning in the German regional, federal state and national levels that illustrate implications of using data of

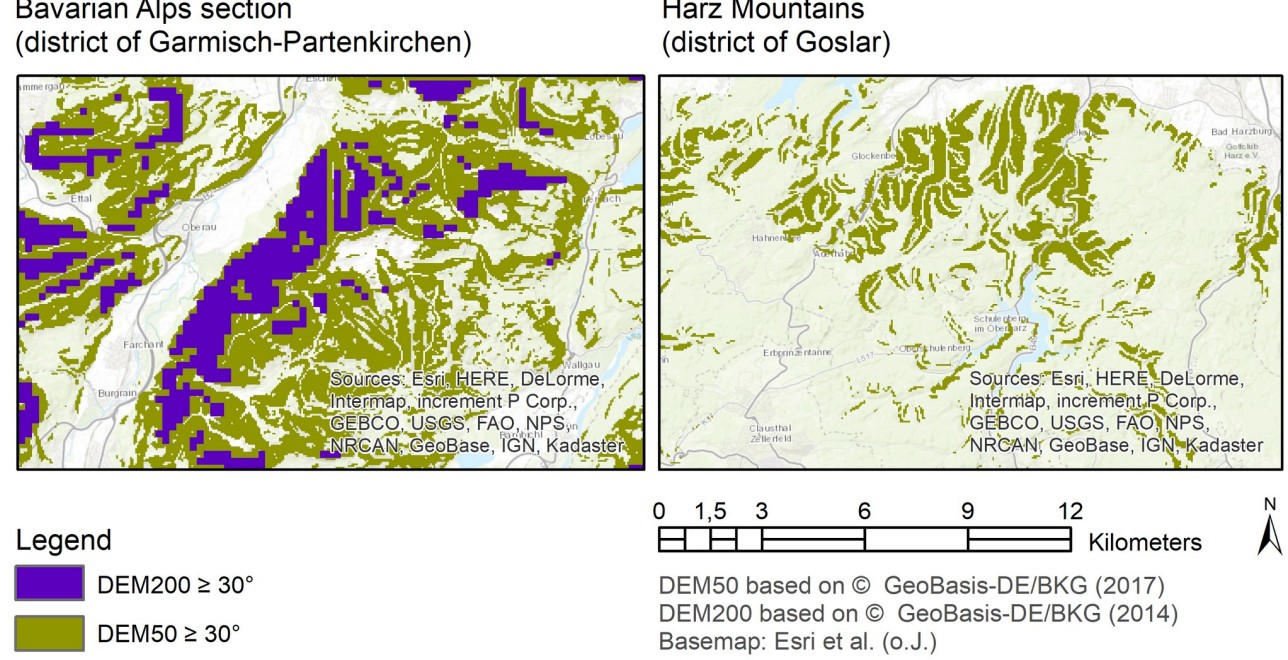

**Fig 5. Comparison of the two elevation models DEM50 and DEM200 with a slope ≥ 30° illustrated by the example of a Bavarian Alps section (left) and a Harz section (right).**

different kinds and levels of uncertainty on environmental assessments and planning recommendations. All findings are discussed in the light of the integration of uncertainty assessments in planning practise still lacking [25, 26].

The case study on 'regional soil-based climate change mitigation' showed that in landscape planning the use of coarse data like CLC or data with fewer classes such as in ATKIS goes hand in hand with a serious loss of valuable information which can lead to a completely different planning statement. Areas with a high relevance for environmental planning are increasingly affected by inconsistencies which translate to uncertainties in potential planning proposals. The case study 'State-wide habitat conservation siting' showed that even the use of high resolution data might lead to erroneous results if aggregated thematic information does not match the classes of interest, as the increase in area with conservation need is mainly due to a lack in specific information in the ATKIS data rendering the distinction between different features as guesswork. Unfortunately, respective areas are of high importance for landscape planning and environmental planning in general. The third case study, 'National renewable energy planning', illustrated that applying coarser vs. more detailed spatial data in efforts to identify area potentials for renewable energy can yield differences by about 2980 $km^2$ in falsely allocated areas for onshore wind energy. The potential areas for onshore wind energy have decreased by 34% when using more detailed data. Taken together, the results re-emphasize the importance of carefully selecting datasets for landscape and environmental analysis for planning, as the first impression of sufficient data certainty might be misleading, and error-propagation through data application in models can result in substantially misleading results. Also more information is needed about characteristics of data sets. We found that data sets may be selective and inhomogeneous in their level of detail which may have considerable influence when used in assessment procedures.

The results of the three illustrative case studies also provide opportunities for reflecting on the implications for planning recommendations. Based on the findings, we distinguish four problem constellations in which the data uncertainty should be considered inappropriate as a basis for decision-support in landscape and environmental planning. The first constellation is when the potential errors in planning recommendations may lead to a pronounced different decision of a relevant dimension which counteracts legitimate planning objectives. This applies, for example, in the case study 'National renewable energy planning' where the results using easily obtained data could not been used for downscaling development limits for renewable energy because false allocations would harm environment and people.

A second problem constellation of inappropriate data uncertainty emerges if potential decisions based on this data lead to consequences which are destructive in a dimension for which the actual decision maker cannot take the responsibility. This applies, for example, if the decisions affect a longer time span than the decision-makers regular phase in office, or if it affects a larger area than that of which she or he has legitimate decision power. This problem constellation is demonstrated in both the case of 'regional soil-based climate change mitigation', where regional decisions about GHG mitigation affect, if so only marginally, the global climate conditions. In this case, the certainty of the assessment results based on ATKIS or even more so CLC 25 data does not provide sufficient certainty to find the areas where climate protection measures would be most needed and efficient and justify mandatory protection and management measures. Many areas of special importance would be overlooked others would be protected unnecessarily. The problem constellation also occurs in the case of 'State-wide habitat conservation siting', when, for example areas of European value are not identified and not appropriately protected by federal state-level decision-makers.

The third problem constellation is represented in situations where consequences of a decision may affect the rights of private subjects in a disproportionate way, in particular to use

ones' own property. Such impacts are usually forbidden, for example by the German constitution. This problem constellation also applies in the case studies of 'regional soil-based climate change mitigation', when the uncertain data would lead to recommendations that force land users to adopt particular GHG retention measures that are not actually helpful based on the real land use of the area. Vice-versa, it applies in the case of 'State wide habitat siting' when farmers are obliged to mow their grasslands only late in the season to protect particular breeding bird species, which in fact do not occur on those conditions.

Finally, the fourth problem constellation is when a large effort or expensive response would be required to address the issues raised by the model outcome and the data uncertainty is also large. Even if no harm would be caused by misallocating the response action, the resulting costs would violate the rules of sound financial management and budgetary rigour that should be applied at all political levels in the European Union. The case study 'regional soil-based climate change mitigation' shows that the certainty of assessment results based on ATKIS or even more so CLC 25 data (easily available data) are not suitable for an appropriate identification of areas where climate protection measures would be most efficient. In all of those problem constellations, planners need to be careful in making decisions and should apply the strategies of caution as outlined in the introduction, for example to try to use more precise data, or to apply the participation, adaptive management or precautionary principle.

We are aware that data uncertainty is only one component of the uncertainty spectrum and we posit that future research is also needed to incorporate considerations of all types of uncertainties in practical planning. In conjunction with our findings we propose that opportunities for future research are to develop robust and pragmatic methods for identifying the uncertainty levels in environmental data and assessment results, and to devise approaches for enhancing data certainty wherever possible. Developments in the remote sensing community on data quality assessment and communication [27, 28] can act as a suitable starting point. Established guidelines on data quality, as used for earth observation data [29], can be easily adapted to be used in data generation in general. In addition, planners should pay much more attention to distortions in modeling results due to data limitations. Uncertainties in assessment results and deduced planning measures have to be interpreted as to their relevance for the issues at stake, leading to more informed decision-making in the face of uncertainty. In many cases, a fast and preliminary decision within bounded uncertainty that addresses the problem constellations above would be better than an ill-informed decision or no decision at all. The introduced problem constellations already cover a broad range of potential planning and decision dilemmas but can only act as a starting point and should be extended when more experiences will be compiled and analyzed. Future research towards this goal should also cover other known types of uncertainties in planning which would only help in tackling these complex problems.

The realistic dilemma of using easily available data instead of better but hard to obtain data with uncertain added value in terms of relevance for answering the planning question will not vanish in the near future. In this situation it is important that uncertainties have to be communicated to the users of planning recommendations. The analysis of uncertainties though, still proposes a multitude of challenges for the practical planner [30]. In order to support practical environmental planning in dealing with uncertainties in a reasonable way, uncertainty analyses with big data sets should be performed for all standard geodata types which are used in environmental planning. This will produce uncertainty markers which can readily be used in practical planning as a basis for calculating the uncertainty in modelling or ecosystem service assessment.

If the uncertainty of the readily available data is known, the uncertainty of the assessment results as well as consequences for deduced respective measures can be judged as to their

acceptability. Furthermore, planning recommendation can be adapted to the uncertain nature of the assessment results. For instance, the comparably little difference between ATKIS and CLC in the overall uncertainty of the state data, can be interpreted as a possibility for using CLC as very actual and readily available data when summarized results are expected for defining general risks or deciding about need for action. However, in most landscape planning applications this cannot be accepted for example for monitoring the development of habitat area or even for roughly calculating the land use induced $CO_2$ emissions on state level. In particular when area specific delimitations are needed and consequences for private property anticipated even ground truth mapping may be required in order to avoid unwanted consequences. Finally, in order to enhance the acceptability of uncertainty communication to decision makers, future research should explore how behavioral interventions such as splitting complex, uncertain decisions into smaller decisions with regard to space and time [31] would increase acceptability.

In a nutshell, the differences detected in the uncertainty of different land use classes were surprisingly high as well as the effect this had on the uncertainty of thematic ES-assessment results and thus on planning recommendations. This emphasizes our plea towards practical planning to pay more attention to data uncertainties as well as to distortions in their propagation in modelling and assessment.

## Acknowledgments

The Authors would like to thank Jordan Siebel for his help in preparing the manuscript. We also thank the editor and two reviewers for their constructive comments on an earlier version of this manuscript.

## Author Contributions

**Conceptualization:** Felix Neuendorf, Christian Albert, Christina von Haaren.

**Data curation:** Felix Neuendorf, Julia Thiele.

**Formal analysis:** Felix Neuendorf, Julia Thiele.

**Investigation:** Julia Thiele.

**Methodology:** Felix Neuendorf.

**Project administration:** Christian Albert, Christina von Haaren.

**Supervision:** Christian Albert, Christina von Haaren.

**Validation:** Julia Thiele, Christian Albert.

**Visualization:** Felix Neuendorf, Christina von Haaren.

**Writing – original draft:** Felix Neuendorf, Julia Thiele, Christina von Haaren.

**Writing – review & editing:** Felix Neuendorf, Julia Thiele, Christian Albert, Christina von Haaren.

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
