## [Decision Letter · Decision Letter 0]

23 May 2021

PONE-D-21-14058

Data uncertainty in LU/LC data is unevenly distributed and has disproportionately high impacts on assessment results in environmental planning

PLOS ONE

Dear Dr. Neuendorf,

Thank you for submitting your manuscript to PLOS ONE. After careful consideration, we feel that it has merit but does not fully meet PLOS ONE’s publication criteria as it currently stands. Therefore, we invite you to submit a revised version of the manuscript that addresses the points raised during the review process.

The study seem to have overly concentrated on spatial resolution as the main source of uncertainty in environmental modelling to the neglect of equally important sources/factors, including from the modelling process itself. Please, provide a justification for this or explore other sources in line with recommendations from the reviewers. In addition, kindly explore possibilities of improving the clarity of the structure and text, as well as strengthening the discussion based on recent works. Please, pay attention to comments on terminologies and ensure consistency. Further, ensure that you have provided and/or corrected the relevant information in the supporting data.

We look forward to receiving your revised manuscript.

Kind regards,

Gerald Forkuor

Academic Editor

PLOS ONE

Journal Requirements:

We note that you have stated that you will provide repository information for your data at acceptance. Should your manuscript be accepted for publication, we will hold it until you provide the relevant accession numbers or DOIs necessary to access your data. If you wish to make changes to your Data Availability statement, please describe these changes in your cover letter and we will update your Data Availability statement to reflect the information you provide.

Reviewers' comments:

Reviewer's Responses to Questions

**Comments to the Author**

1. Is the manuscript technically sound, and do the data support the conclusions?

Reviewer #1: Yes

Reviewer #2: Partly

2. Has the statistical analysis been performed appropriately and rigorously? 

Reviewer #1: Yes

Reviewer #2: No

3. Have the authors made all data underlying the findings in their manuscript fully available?

Reviewer #1: No

Reviewer #2: No

4. Is the manuscript presented in an intelligible fashion and written in standard English?

Reviewer #1: Yes

Reviewer #2: Yes

5. Review Comments to the Author

Reviewer #1: General comments:

In this manuscript, the authors assess the implications of data uncertainty on assessments in environmental planning based on two practical case studies. Uncertainties are important but often ignored in environmental assessments, so this manuscript addresses a relevant gap in the literature. However, I find that the clarity of both the structure and text could be improved, and the discussion could be strengthened with reference to existing work on uncertainties in general and more specific data uncertainties in spatial data. In addition, I think a stronger focus on the criteria for when the uncertainty is too high for specific applications would highlight the novelty of this paper. I would also suggest a language edit, as there are several typos and unclear formulations (see details in specific comments below).

1) In this manuscript, the authors mainly address data uncertainty. In particular, they focus on thematic accuracy and spatial resolution. While these are important sources of uncertainty, it would be helpful to put them in context with other types of uncertainties that affect assessments and decisions in environmental planning (such as model uncertainties, the inherent variability of environmental systems, or ambiguity among views of different stakeholders). How important are data uncertainties in relation to other sources of uncertainty? Addressing this question would add some depth to the discussion.

2) A clearer structure would help to make the paper easier to follow. In particular, I think it would be important to clearly define the criteria of when uncertainty is “too high” at an early stage in the manuscript, as this is a particularly novel and relevant aspect. Then, the results could be presented in relation to these criteria. Furthermore, I would suggest organizing the Methods section by case studies, and clearly describing the specific questions (as currently described in the discussion), data, and type of uncertainty addressed in each case study. It might also be helpful to have a short reference for each case study, e.g. Case study A and B or “carbon mapping” and “renewable energies”, which would be used consistently throughout the paper.

3) It seems to me that especially the results of the second case study (choice of suitable areas for RE) mainly highlight the need to select the appropriate spatial and thematic resolution of data, rather than the uncertainty of the data itself. The importance of choosing the appropriate resolution is well known in the spatial modelling/assessment community, but its implications for practical decisions in environmental planning are still interesting. I think it would therefore be useful to explicitly state the types of data uncertainty that are addressed in each case study (e.g. misclassification, loss of information due to spatial or thematic aggregation and insufficient resolution, etc.).

4) There has been a lot of work in the remote sensing community about various dimensions and metrics of data quality, and their effect on the usability of geospatial data. I think that referring to this literature would help strengthen the content of this paper, see e.g.:

• GEOSS Data Quality guidelines (https://www.earthobservations.org/documents/dsp/GEOSS_Data_Quality_Guidelines.pdf)

• Batini C, Blaschke T, Lang S, Albrecht F, Abdulmutalib HM, Barsi, et al. Data quality in remote sensing. International Archives of the Photogrammetry, Remote Sensing and Spatial Information Sciences - ISPRS Archives. 2017;42(2W7):447–53.

Specific comments

5) Lines 44-46: “Often…” this sentence is overly complex and difficult to understand, please consider rephrasing.

6) Lines 54-55: It would be helpful to define what authors mean by terms such as “data quality” and “preciseness”, as data quality includes both accuracy and precision, as well as some other attributes (e.g. appropriate temporal, spatial, and thematic resolution).

7) Line 100: An “error matrix” is commonly called a “confusion matrix” in the field of remote sensing classifications, which are a common source of LU/LC data, so it might be useful to mention both terms here.

8) Table 1: To make it easier to connect this table to the case studies, please consider adding a column or label to indicate which datasets are used for which case study.

9) Table 1 footnote: Typo “statesuse”. What do you mean by topicality in this context?

10) Lines 139-141: I don’t understand this sentence – why are biotope mapping useless because landscape planners are obliged to use appropriate data? Please clarify.

11) Lines 170-171: This sentence seems like it would fit more into the introduction, as it stresses the general relevance of choosing the appropriate spatial and thematic resolution.

12) Line 209: By accuracy, do you mean producer’s or users’ accuracy? Please specify.

13) Line 243: Phrases such as “dramatic effect” would be more appropriate in the discussion.

14) Lines 264-268: These results highlight the need to choose an appropriate spatial resolution, but do not really inform about uncertainty.

15) Lines 281-282: “When looking at…” this sentence is strangely formulated, please consider rephrasing.

16) Line 284: Typo “compatbible” should be “compatible”.

17) Lines 294-295: Here, you might want to mention that certain land-cover classifications (e.g. random forest-based classifications) can in fact provide spatially explicit information about uncertainties.

18) Lines 315-321: These criteria are very relevant. For a clearer structure, I would suggest that these criteria already be mentioned earlier in the manuscript (e.g. in the introduction or methods section), so that the reader can keep them in mind when presented with the results.

19) Lines 322-325: These questions should be presented when the case study in introduced in the methods section.

Reviewer #2: This paper addresses an important topic of data spatial scale and its influence on landscape modeling. The paper shows that using data of different spatial scales (and from different sources) can result in different results when used in modeling. The general conclusion that uncertainty in results is often overlooked in modeling studies is an important one.

Below I list some major concerns. I think a simple solution to addressing my concerns would be to change the theme of the paper away from providing a method of characterizing uncertainty (which the paper does not explore), and instead focus on a comparison of the dependence of modeling results on the spatial resolution and other properties of the input data. I think the latter approach will allow you to still achieve your main aim of highlighting the importance of considering how the data used affects the resulting model.

Major concerns

1. The paper makes broad and sweeping claims, including that it provides a simple and practical method for characterizing uncertainty, that are not supported by the analysis or the results. There is no investigation of the uncertainty of any of the data sets, or in the modeling itself. Also, as a proposed method for characterizing data suitability, by comparing the results with higher spatial resolution data, it does not seem to be useful to me, because why not use the higher spatial resolution data in the first place?

2. There are multiple sources of uncertainty in comparing data sets – spatial resolution is simply one of the sources of differences. Even something as subtle as the difference in how the classes are defined can be important. Other issues include spatial location accuracy, thematic classification accuracy, date of acquisition, etc. The mismatch between the classes required by the model and the classes in the geospatial data may be even greater than the issue of the spatial resolution.

3. After making the correct statement, that “There is no spatial environmental planning without uncertainties”, the paper then goes on to treat the biotype mapping as “ground truth”, and to call differences between the biotype data and other data sets as uncertainty in the other data, not in the biotype data. I think it is a fundamental flaw to suggest the biotype data are the truth and by implication so are the model results using the biotype data.

4. It would be useful to provide a great deal more information on how each data set was generated, and to explain the methods of analysis in more detail. In particular, information on the biotype data seems important – what scale is it produced at and more detail on how it is mapped? How were the DEM products generated (e.g. photogrammetry, lidar, etc), and what is their source? In Table 1, you could separate out the minimum mapping unit area and the pixel size for raster data sets. What is the spatial uncertainty in the location of the data? For the data overlay for generating the confusion matrix, it is useful to state the resampling method used (presumably nearest neighbor). What is the cross-walk for the conversion of the data legends between the different data sets? The concept of the emission level classes (figure 4) and their definitions should be explained in the methods.

5. There appear to be some errors in the supporting document’s calculation of the confusion matrices user’s accuracies. For example, for Saxony Anhalt, I get different values for both Mining and landfill sites (I get 0.0) as well as Areas of sparse vegetation (I get 0.06) user’s accuracies. I suggest checking all the numbers.

6. I suggest presenting both user’s and producer’s accuracies in Figure 2 and the text, and not just user’s accuracies, since producer’s accuracies are equally important. (Though see comments below about the term “accuracy”.)

Minor issues

1. As explained above, I don’t think it is conceptually correct to refer to the differences in the maps as “accuracy” or even “uncertainty”. A better term would be agreement or consistency. For example, you could use the terms “user’s consistency” and “producer’s consistency” instead of “user’s accuracy” and “producer’s accuracy” – though you’d need to define the terms since they are not commonly used. Using agreement or consistency would not preclude making an argument that one data set more closely matches the assumed spatial scale (or other properties) assumed by the model. For example, if the model has wetlands as a key class, and wetlands are generally smaller than the 25 ha MMU of the Corine landcover, then the use of the finer spatial scale data will allow the model to incorporate these smaller wetlands.

2. Avoid “error matrix” for the above reason and instead use a term like “confusion matrix”.

3. Avoid the term “ground truth” – this is generally thought not to be a good term in remote sensing, and in a paper on uncertainty in data layers it seems strange to imply that any data could be the “truth”.

4. I suggest reserving “significant” for “statistically significant”. Instead, for general use, use terms like notable.

5. L28. I don’t see an analysis of the propagation of inaccuracy.

6. L30. I don’t see any “differences in uncertainty”, but instead differences in results.

7. L31. I think you are promising too much in claiming to have demonstrated uneven distribution in uncertainties. However, you have shown that the magnitude of the differences in results between using different data sets can vary, depending on the application and even location.

8. Use a subscript for the 2 in CO2 throughout.

9. L92 – I can’t understand this sentence – perhaps rephrase?

10. L115 – “no significant missing data” – does this mean there is some missing data? If so, please explain.

11. Figure 1. Please add (a) scale bar (in m, or km, as may be appropriate) and (b) a legend.

12. I think it would be useful to point out to the reader that the issue is not just the pixel size of raster data, but the minimum mapping unit area – it is convenient that the CORINE data specifies this so clearly.

13. L189, 190 and 195/196. The details of the grid cell size should be in Table 1. (I suggest using “grid cell size” or “grid cell dimension”, or even “pixel size”, not “grid width”)

14. L195 – In my experience, even 50 m is still very coarse for this type of modeling. I think this reinforces the idea that often in modeling we have to use data that don’t entirely fit our purpose.

15. L198 – “user’s accuracy”, not “user accuracy” (but seem my comment above about not using the term accuracy).

16. L199 – what is “combined accuracy” – do you just mean just “the range in the user’s accuracy”? – but that doesn’t make sense since the lowest value seems to be 0%, not 82%? Do you perhaps mean the “overall accuracy” (which is defined as the sum of the correctly classified areas in each class, divided by the total area of the map)?

17. Figure 2. The y-axis should be “Agreement”, not “Spatial accuracy” (the latter would be a measure of the spatial registration of the two data sets, which I don’t believe is what you mean). Is it possible to use color – I struggled to tell the different grays apart.

18. L219 – “correlation” – do you perhaps mean “confusion”?

19. Figure 3. Are the legends correct? For example, for Figure 3 (a), my interpretation from the legend (“misclassified GRASSLAND”) is that these are areas that are in reality GRASSLAND but have been misclassified. But I think you actually might mean “Proportion of pixels misclassified as GRASSLAND”?

20. L260 “rundue” should be two words.

21. L274 – replace “geometric” with “spatial resolution”.

22. L294/295 – I don’t understand the sentence (or the subsequent discussion) and my best interpretation doesn’t seem to be correct. The confusion matrix is a spatial overlay and produces a summary of agreement. It is true however that it does not consider the contiguity of the distribution of pixels in any one class, but I don’t see why that should matter.

23. L300 – yes, it has uncertainty. Since “to a certain degree” are not defined, they can be removed.

24. L302 - The term “geodata uncertainty marker” is not defined and it is not clear any case that one can generalize for other applications – your work shows that even for different areas using the same dataset, the agreement varies. I suggest dropping this.

25. L321 – I don’t understand why you are saying “the uncertainty is too high” when the “effect is small”?

26. L332 I don’t understand why small land holders would be affected more than large ones. Wouldn’t small ones be too small to show up in the analysis, and therefore be ignored? Perhaps you could elaborate.

27. L345. I suggest not using a term like “better” (which is not defined), but instead simply state the properties that you mean by “better” – e.g. data with a spatial resolution that matches the modeling requirements, or whatever you mean.

28. L359. I don’t think you have analyzed the uncertainty in the products you have generated, as there are many other sources of uncertainty, including in the model itself, not to mention uncertainty in the “ground truth” data. I think instead you have shown that different data sets can produce different results when applied to a model, and therefore it is important to consider what spatial scale (as well as other data characteristics, such as classes mapped, etc) the model assumes for input data.

29. Figure 4. Could you use color here, too, to make it easier to interpret?

30. Supporting data. What are the units of measurement in the supporting confusion matrix data tables? Ha, pixels, etc? (if pixels, what size?). The tables are very hard to read because of the decimals (it is hard to get a visual idea of where the confusions lie) and the size of the numbers. Units like square km would make the table a lot more visually understandable, and perhaps make your overall points easier to make.

31. Supporting data. Very important – all the user’s and producer’s accuracies don’t seem to be %, though they are labeled as such. I think they are just straight proportions (on a 0-1 scale).

32. Supporting data – what does “balance" mean? What is "sum/balance"?

6. PLOS authors have the option to publish the peer review history of their article (what does this mean?). If published, this will include your full peer review and any attached files.

Reviewer #1: **Yes: **Ana Stritih

Reviewer #2: No

---

## [Author Response · Author response to Decision Letter 0]

4 Aug 2021

Editor comments 

The study seem to have overly concentrated on spatial resolution as the main source of uncertainty in environmental modelling to the neglect of equally important sources/factors, including from the modelling process itself. Please, provide a justification for this or explore other sources in line with recommendations from the reviewers. 

 Response: Dear Editor, Dear Dr. Forkuor, thank you for your in-depth review and the critical remark. We understand that there, we obviously did not manage to express ourselves clear enough and have now revised the manuscript accordingly.

We aknowledge the fact that there are multiple sources of uncertainty. For that we added a whole section to the introduction highlighting different sources of uncertainties in planning (Lines 37-56)

To highlight the importance of data uncertainty we write: 

“Uncertainty of topographical data presents one of the most important uncertainties, as such input data inaccuracies propagates through the entire environmental assessment and planning process. Relevant concerns of environmental planning in terms of data uncertainties include, among others, mistakes in categorising (for example, if particular land parcels are incorrectly attributed to land use types), inconsistencies between the different datasets (mainly the location and spatial delineation of features) and impacts of data resolution (for instance if the resolution it too coarse to divide site-specific management recommendations to farmers). Some of these uncertainties may be obvious, but not quantified, others may be hidden in the data set. Both types of uncertainties are usually not explicitly communicated with the data provision. Nevertheless, the planner has to decide which data detail will suffice for which specific planning task – risking misjudgement.” (Lines 57-66)

In addition, we write in the discussion section that we are fully aware of the fact that other, additional sources of uncertainty also play a role, but that it was beyond the scope of this paper to explore them in greater detail. We write:

“We are aware that data uncertainty is only one component of the uncertainty spectrum and we posit that future research is also needed to incorporate considerations of all types of uncertainties in practical planning. In conjunction with our findings we propose that opportunities for future research are to develop robust and pragmatic methods for identifying the uncertainty levels in environmental data and assessment results, and to devise approaches for enhancing data certainty wherever possible.” (Lines 366-371)

In addition, kindly explore possibilities of improving the clarity of the structure and text, as well as strengthening the discussion based on recent works. Response: Thanks. We have have done our best to improve the clarity of the manuscript structure by revisiting the writing of the entire manuscript and adapting it to increase the flow of argumentation and enhance the style of writing. We have done so by critically re-reading and carefully editing the entire manuscript ourselves, and we have asked a language consultant to review and improve the manuscript for us.

In addition, we have strengthened the discussion by better weaving our arguments into the latest state of knowledge as reflected in recent publications. 

Please, pay attention to comments on terminologies and ensure consistency. Further, ensure that you have provided and/or corrected the relevant information in the supporting data. Thanks for flagging this. We now consistently refer to data uncertainty in the manuscript when referring to the kind of uncertaintywe have considered. The provided supporting data is no longer relevant for the manuscript and has been removed

 Response: Thanks – yes, we have done as suggested.

2) We note that you have stated that you will provide repository information for your data at acceptance. Should your manuscript be accepted for publication, we will hold it until you provide the relevant accession numbers or DOIs necessary to access your data. If you wish to make changes to your Data Availability statement, please describe these changes in your cover letter and we will update your Data Availability statement to reflect the information you provide. Response: Thank you for noting this. We are in the process of making the data publicly available on our own repository. The access will be provided via DOI numbers.

The data for the third case study is already available, the data for case study one and two will be available in short time.

Reviewer #1 

GENERAL COMMENTS: 

In this manuscript, the authors assess the implications of data uncertainty on assessments in environmental planning based on two practical case studies. Uncertainties are important but often ignored in environmental assessments, so this manuscript addresses a relevant gap in the literature. Response: Thank you very much for the insightful review and the many comments for further improvement. Please excuse the short handling of some of your specific comments as these have been obsolete after the complete revision of the manuscript. Nonetheless we would like to thank you for each comment as these greatly helped us in improving the whole manuscript.

However, I find that the clarity of both the structure and text could be improved, and the discussion could be strengthened with reference to existing work on uncertainties in general and more specific data uncertainties in spatial data. Thanks for noting. We agree and have substantially and comprehensively revised the entire manuscript. In fact, in order to comply with your suggestions, we needed to revise everything as you will see in the track-changes version. We hope this meets your expectations. 

In addition, I think a stronger focus on the criteria for when the uncertainty is too high for specific applications would highlight the novelty of this paper. Response: Thank you for pointing this out. We had a long discussion within the team of authors of how to best address this. We have considered moving the criteira to the introduction or the methods section and tried our best to revise the manuscript accordingly. In the end, however, we realized that the best place to put them is in the discussion where we critically reflect on the case studies. We hope you agree and appreciate the changes. In any way, we are grateful for your advice and suggestions which has spurred our further reflection, and we are confident that this change has substantially enhanced the quality of our manuscript.

I would also suggest a language edit, as there are several typos and unclear formulations (see details in specific comments below). Response: Thank you for the suggestion. A comprehensive language edit has been performed, which we believe has resulted in a major improvement of the quality of our manuscript. 

1) In this manuscript, the authors mainly address data uncertainty. In particular, they focus on thematic accuracy and spatial resolution. While these are important sources of uncertainty, it would be helpful to put them in context with other types of uncertainties that affect assessments and decisions in environmental planning (such as model uncertainties, the inherent variability of environmental systems, or ambiguity among views of different stakeholders). How important are data uncertainties in relation to other sources of uncertainty? Addressing this question would add some depth to the discussion. Response: Thank you for this comment. Data is indeed only a small part of the uncertainty problem. We rephrased the introduction and included a broader overview of uncertainties in environmental planning. 

“A recent review revealed that the main planning relevant categories of uncertainties in environmental planning can be associated with four categories [3]: Data uncertainties, which are most commonly associated with geometric uncertainties that arise mainly from resolution, thematic uncertainties that occur when values are attributed to objects, and temporal mismatches between data sources [4]. Model uncertainties, the second category, relate to challenges in choosing appropriate indicators and variables for environmental assessments [5]. Models applying only a small number of explanatory variables are generally easier to use for planners, but are often inferior to more complex and dynamic models in understanding environmental system change. The most obvious trade-off in model development for planning purposes is between precision and usefulness, for example when a complex model would become more demanding in data requirements [6]. In addition, a very high level of detail can be impractical and a source of uncertainty in planning applications when the interpretation of complex model results puts high demands on user [7]. The third category, projection uncertainties, is linked to projecting future states of the environment � atypical challenge for planning as a strong forward looking action [8]. Plans often suggest that problems are of a static nature, but in reality they are highly dynamic [9] and can often not be predicted with the understanding of future behaviour of various components [10]. Finally, valuation uncertainties are introduced when norms and laws need to be interpreted to form scales for valuation [3]. While data, model and valuation uncertainties are prominent at multiple stages of the planning cycle, projection uncertainties come into play in particular when scenarios for future developments are created and potential scenario impacts are being explored.” (Lines 37-56

We added a new part that highlights the importance of data uncertainty:

“Uncertainty of topographical data presents one of the most important uncertainties, as such input data inaccuracies propagate through the entire environmental assessment and planning process. Relevant concerns of environmental planning in terms of data uncertainties include, among others, mistakes in categorising (for example, if particular land parcels are incorrectly attributed to land use types), inconsistencies between the different datasets (mainly the location and spatial delineation of features) and impacts of data resolution (for instance if the resolution it too coarse to divide site-specific management recommendations to farmers). Some of these uncertainties may be obvious, but not quantified, others may be hidden in the data set. Both types of uncertainties are usually not explicitly communicated with the data provision. Nevertheless, the planner has to decide which data detail will suffice for which specific planning task – risking misjudgement.” (Lines57-66)

We pick up on that briefly in the discussion but focused on our findings and possible implications:

“We are aware that data uncertainty is only one component of the uncertainty spectrum and we posit that future research is also needed to incorporate considerations of all types of uncertainties in practical planning. In conjunction with our findings we propose that opportunities for future research are to develop robust and pragmatic methods for identifying the uncertainty levels in environmental data and assessment results, and to devise approaches for enhancing data certainty wherever possible.” (Lines 366-371)

2) A clearer structure would help to make the paper easier to follow. In particular, I think it would be important to clearly define the criteria of when uncertainty is “too high” at an early stage in the manuscript, as this is a particularly novel and relevant aspect. Then, the results could be presented in relation to these criteria. Furthermore, I would suggest organizing the Methods section by case studies, and clearly describing the specific questions (as currently described in the discussion), data, and type of uncertainty addressed in each case study. It might also be helpful to have a short reference for each case study, e.g. Case study A and B or “carbon mapping” and “renewable energies”, which would be used consistently throughout the paper. Response: Dear reviewer, thank you for pointing this out. We fully agree that the structure of our manuscript could be improved. We have invested substantial time and efforts over the last weeks to address this issue, including several rounds of discussion and reflection within the author teams, and the elaboration of several versions and improvements of the manuscript. 

In this process, we have come to the conclusion that the criteria for deciding about the level at which uncertainties are too high are not so easy to draw generically, but that this rather requires a case-specific contemplation and consideration. As mentioned, explained and justified above, we have decided to move the criteria into the discussion.

Our new manuscript outline takes up your kind advance to structure the methods section by case studies. As suggested by you, we now clearly descibre each specific question, we state the kind of data we use and our estimation of the inherent data uncertainty. We also describe that we define two scenarios for each case study, with the use of different levels of data uncertainties, and we clarify how we derive planning recommendations for these scenarios. 

We also follow your advice about the naming of the case studies: They are now referred to as “Regional soil-based climate change mitigation”, “State-wide habitat conservation siting”, and “National renewable energy planning”

3) It seems to me that especially the results of the second case study (choice of suitable areas for RE) mainly highlight the need to select the appropriate spatial and thematic resolution of data, rather than the uncertainty of the data itself. The importance of choosing the appropriate resolution is well known in the spatial modelling/assessment community, but its implications for practical decisions in environmental planning are still interesting. I think it would therefore be useful to explicitly state the types of data uncertainty that are addressed in each case study (e.g. misclassification, loss of information due to spatial or thematic aggregation and insufficient resolution, etc.). Response: We agree with you that with the second (now third) case study we mainly considered the resolution of the spatial input data in a nationwide assessment of human- and nature-compatible usablearea potentials for renewable energies (RE) and not the uncertainty (e.g. misclassification) of the spatial input data itself.With this study, we were able to emphasize that high-resolution input data should be used, even for studies on a nationwide scale, in order to increase the quality of the resulting statementsor recommendations which is more in line with the updated focus of the paper. Future studies in the field of potential area assessmentsof RE can now refer to this study.

In order to clarify this, we have adjusted the manuscript:

“and impacts of data resolution” (Addition in the introduction) (Line 62)

“Case study number three, ‘National renewable energy planning’, concerns an environmental planning effort aiming to identify areas for the siting of renewable energies, which should also guide decision-making at the regional level.” (Lines 110-112)

 “Two model runs were performed with input data of different resolutions to quantify the influence of data resolution to identify opportunity spaces for generating renewable energy, taking into account criteria for human and nature conservation compatibility.” (Lines 189-191)

4) There has been a lot of work in the remote sensing community about various dimensions and metrics of data quality, and their effect on the usability of geospatial data. I think that referring to this literature would help strengthen the content of this paper, see e.g.:

• GEOSS Data Quality guidelines (https://www.earthobservations.org/documents/dsp/GEOSS_Data_Quality_Guidelines.pdf)

• Batini C, Blaschke T, Lang S, Albrecht F, Abdulmutalib HM, Barsi, et al. Data quality in remote sensing. International Archives of the Photogrammetry, Remote Sensing and Spatial Information Sciences - ISPRS Archives. 2017;42(2W7):447–53. Response: Thank you for this important comment. We agree with you that these publications and the work in the remote sensing community would have strengthend the core aspect of the first version of the paper. Since the focus is different now, we opted to include these references in the discussion as a starting point for the planner community to build upon. The lines read as follows:

‘Good starting points are developments in the remote sensing community on data quality assessment and communication (25, 26). Established guidelines on data quality, as used for earth observation data [27], have the potential to be used in data generation in general.’ (Lines 368-371)

SPECIFIC COMMENTS:

5) Lines 44-46: “Often…” this sentence is overly complex and difficult to understand, please consider rephrasing. Response: This part of the manuscript has been completely revised.

6) Lines 54-55: It would be helpful to define what authors mean by terms such as “data quality” and “preciseness”, as data quality includes both accuracy and precision, as well as some other attributes (e.g. appropriate temporal, spatial, and thematic resolution). Response: This part of the manuscript has been completely revised.

7) Line 100: An “error matrix” is commonly called a “confusion matrix” in the field of remote sensing classifications, which are a common source of LU/LC data, so it might be useful to mention both terms here. Response: This part of the manuscript has been completely revised.

8) Table 1: To make it easier to connect this table to the case studies, please consider adding a column or label to indicate which datasets are used for which case study. Response: Thank you for this comment, we added a column with information to the respective case study.

9) Table 1 footnote: Typo “statesuse”. What do you mean by topicality in this context? Response: Topicality means actuality in this context. We aknowledge that it might be a misleading term, so we changed it to actuality instead.

10) Lines 139-141: I don’t understand this sentence – why are biotope mapping useless because landscape planners are obliged to use appropriate data? Please clarify. Response: This part of the manuscript has been completely revised.

11) Lines 170-171: This sentence seems like it would fit more into the introduction, as it stresses the general relevance of choosing the appropriate spatial and thematic resolution. Response: This part of the manuscript has been completely revised.

12) Line 209: By accuracy, do you mean producer’s or users’ accuracy? Please specify. Response: This part of the manuscript has been completely revised.

13) Line 243: Phrases such as “dramatic effect” would be more appropriate in the discussion. Response: Thank you for this remark, we carefully checked the new manuscript for such phrases and made sure they were located correctly.

14) Lines 264-268: These results highlight the need to choose an appropriate spatial resolution, but do not really inform about uncertainty. Response: Absolutely, therefore we have included some adjustments into the manuscript to clarify this. We have listed the adjustment in the response to your third major concern ("It seems to me that..."); thank you very much for your comment.

15) Lines 281-282: “When looking at…” this sentence is strangely formulated, please consider rephrasing. Response: This part of the manuscript has been completely revised.

16) Line 284: Typo “compatbible” should be “compatible”. Response: Thank you, we thoroughly checked the manuscript for spelling errors.

17) Lines 294-295: Here, you might want to mention that certain land-cover classifications (e.g. random forest-based classifications) can in fact provide spatially explicit information about uncertainties. Response: This part of the manuscript has been completely revised.

18) Lines 315-321: These criteria are very relevant. For a clearer structure, I would suggest that these criteria already be mentioned earlier in the manuscript (e.g. in the introduction or methods section), so that the reader can keep them in mind when presented with the results. Response: We discussed about reframing the manuscript and putting the criterias at the front but opted against it. The criterias have been reformulated into problem constellations and discussed thoroughly in the discussion and conclusion section.

19) Lines 322-325: These questions should be presented when the case study in introduced in the methods section. Response: This part of the manuscript has been completely revised.

Reviewer #2 

GENERAL COMMENTS:

This paper addresses an important topic of data spatial scale and its influence on landscape modeling. The paper shows that using data of different spatial scales (and from different sources) can result in different results when used in modeling. The general conclusion that uncertainty in results is often overlooked in modeling studies is an important one. Response: Dear Reviewer, thank you very much for these important remarks. The revised manuscript now picks up on this and goes much more in depth.

Please excuse the short handling of the majority of your minor remarks, but most of them are reffering to text bodies and subjects that no longer are a focal point in the manuscript.

Below I list some major concerns. I think a simple solution to addressing my concerns would be to change the theme of the paper away from providing a method of characterizing uncertainty (which the paper does not explore), and instead focus on a comparison of the dependence of modeling results on the spatial resolution and other properties of the input data. I think the latter approach will allow you to still achieve your main aim of highlighting the importance of considering how the data used affects the resulting model. Response: Thanks for suggesting this solution, which we very much appreciate and follow in our revised manuscript.

We now exlusively focus on a case study-based exploration of the impacts of data uncertainty on planning recommendations. We descreibt the methods applied in each of the three case studies in detail in the methods section, and we clearly present the results for each case study in the results section. We thereby distinguish two scenarios in each case study – one scenario applying data of very high uncertaintiy levels, and one scenario applying data with lower uncertainty. The comparision of recommendations deriverd based on results in both scenarios then allow us in the discussion section to critically reflect upon the implications of data uncertainties on environemtnal planning recommendations

MAJOR CONCERNS

1. The paper makes broad and sweeping claims, including that it provides a simple and practical method for characterizing uncertainty, that are not supported by the analysis or the results. There is no investigation of the uncertainty of any of the data sets, or in the modeling itself. Also, as a proposed method for characterizing data suitability, by comparing the results with higher spatial resolution data, it does not seem to be useful to me, because why not use the higher spatial resolution data in the first place? Response: Dear reviewer – thanks for your critical comment which has helped us substantially in reconceptualizting the main aims and the entire concept of the paper. 

Regarding your question about why not using better data in the first place, we would like to argue that the main idea of the paper is now to illustrate the implications of applying data of difererent levels of uncertaintiy on planning recommendations. We follow this aim because environmental planners always need to make planning recommendations amid uncertainties and have devised some simple heuristics, i.e. planning principles, for intuitively addressing uncertainties. However, we find that this can cause serious harm, as planners often cannot appropriately estimate the degree of data uncertainty truly involved in their assessments. Thus, we wrote this paper with the intention to shed light on the implications of applying data with different uncertainty levels in decision making. In fact, planners often lack the appropriate certain data, so our cases are just for illustration of what differences using this sub-optimal data can make.

2. There are multiple sources of uncertainty in comparing data sets – spatial resolution is simply one of the sources of differences. Even something as subtle as the difference in how the classes are defined can be important. Other issues include spatial location accuracy, thematic classification accuracy, date of acquisition, etc. The mismatch between the classes required by the model and the classes in the geospatial data may be even greater than the issue of the spatial resolution. Response: We aknowledge the fact that there are multiple layers to data uncertainty and uncertainty in general, thank you for this important remark. We added some parts to the introduction and the discussion to refer to other sources of uncertainty in planning. To cover other data related uncertainties such as temporal factors we chose the data that best matched the other data we used, for example on crops, which resembles the work actual planners should have done.

3. After making the correct statement, that “There is no spatial environmental planning without uncertainties”, the paper then goes on to treat the biotype mapping as “ground truth”, and to call differences between the biotype data and other data sets as uncertainty in the other data, not in the biotype data. I think it is a fundamental flaw to suggest the biotype data are the truth and by implication so are the model results using the biotype data. Response: Thank you for this very important comment. We aknowledge that Ground truth is a misleading term in this case. We opted to substitute the term “ground truth” with “reference” and changed this wording for the manuscript. Nevertheless we still believe that the biotope mapping is the best data available in our case and that there is no inherent flaw in using it as reference data. This data simply has the highest thematic and spatial accuracy by a great margin. We agree that there are also uncertainties inherent to this data, and added this to the data description section. This also correlates with your next important comment.

4. It would be useful to provide a great deal more information on how each data set was generated, and to explain the methods of analysis in more detail. In particular, information on the biotype data seems important – what scale is it produced at and more detail on how it is mapped? How were the DEM products generated (e.g. photogrammetry, lidar, etc), and what is their source? In Table 1, you could separate out the minimum mapping unit area and the pixel size for raster data sets. What is the spatial uncertainty in the location of the data? For the data overlay for generating the confusion matrix, it is useful to state the resampling method used (presumably nearest neighbor). What is the cross-walk for the conversion of the data legends between the different data sets? The concept of the emission level classes (figure 4) and their definitions should be explained in the methods. Response: Thank you for this important comment. We reworked the section about the different data sets and included more information on how the data was collected and what the minimum mapping unit where. It should now give a better understanding of the different data sets. We also added information to the table that gives an overview of the different data sets and hope that this is now more comprehensible. 

We also added a sentence regarding the emission level classes and added respective information to the new results table for case study one. We aknowledge the fact that the model chosen is important in general but for our case it was only important to use a method that takes all assessed data sets as input. We expect that all methods that address emission and retention would be affected by changes in extremely important classes, like moors and heathlands. 

“For each dataset, we applied the same method for estimating potential CO2-emission and retention potentials [19]. The applied method combines data about soil types and land-use/land-cover which are classified as to climate relevance, and should be available for a coherent time span of 5 years.” (Lines 168-171; Table 2)

5. There appear to be some errors in the supporting document’s calculation of the confusion matrices user’s accuracies. For example, for Saxony Anhalt, I get different values for both Mining and landfill sites (I get 0.0) as well as Areas of sparse vegetation (I get 0.06) user’s accuracies. I suggest checking all the numbers. Response: Thank you for pointing this out, there has indeed been an error in the calculation of several classes.The supporting information in question though is no longer relevant for the paper and has been removed. 

6. I suggest presenting both user’s and producer’s accuracies in Figure 2 and the text, and not just user’s accuracies, since producer’s accuracies are equally important. (Though see comments below about the term “accuracy”.) Response: We agree that it would be beneficial to present both accuracies, but due to a new focus and structure the confusion matrix is no longer part of the results.

MINOR ISSUES:

1. As explained above, I don’t think it is conceptually correct to refer to the differences in the maps as “accuracy” or even “uncertainty”. A better term would be agreement or consistency. For example, you could use the terms “user’s consistency” and “producer’s consistency” instead of “user’s accuracy” and “producer’s accuracy” – though you’d need to define the terms since they are not commonly used. Using agreement or consistency would not preclude making an argument that one data set more closely matches the assumed spatial scale (or other properties) assumed by the model. For example, if the model has wetlands as a key class, and wetlands are generally smaller than the 25 ha MMU of the Corine landcover, then the use of the finer spatial scale data will allow the model to incorporate these smaller wetlands. Response: Thank you for this insightful comment, we fully agree that definitions of terms are of great importance. Our new manuscript version now includes the addition of various kinds of uncertainty which are common in environmental planning and and explanation that we focus exclusively on data uncertainty and how it depicts in planning results. Since the whole focus has been shifted, the confusion matrix is no longer a focal point of the analysis, albeit the basic principle of overlaying data still remains. In these cases we use the term consistency for the manuscript. We also introduced that term in the introduction. The section reads as follows:

“Relevant concerns of environmental planning in terms of data uncertainties include, among others, mistakes in categorising (for example, if particular land parcels are incorrectly attributed to land use types), inconsistencies between the different datasets (mainly the location and spatial delineation of features) and impacts of data resolution (for instance if the resolution it too coarse to divide site-specific management recommendations to farmers).” (Lines 58-63)

2. Avoid “error matrix” for the above reason and instead use a term like “confusion matrix”. Response: Thank you for this comment, as already mentioned, the confusion matrix is no longer part of the manuscript.

3. Avoid the term “ground truth” – this is generally thought not to be a good term in remote sensing, and in a paper on uncertainty in data layers it seems strange to imply that any data could be the “truth”. Response: Thank you for this remark, we changed the wording to “reference” throughout the manuscript to avoid any confusion.

4. I suggest reserving “significant” for “statistically significant”. Instead, for general use, use terms like notable. Response: Thank you for this comment, we changed the wording accordingly.

5. L28. I don’t see an analysis of the propagation of inaccuracy. Response: Thank you for this comment. This statement was not fitting in the abstract and has been completely removed during the rephrasing of that section. We now phrased the aim of this paper as follows:

“This paper aims to explore how different data uncertainties, usually unknown to the planner, may influence environmental planning recommendations.” (Lines 18-20)

6. L30. I don’t see any “differences in uncertainty”, but instead differences in results. Response: This part of the manuscript has been completely revised.

7. L31. I think you are promising too much in claiming to have demonstrated uneven distribution in uncertainties. However, you have shown that the magnitude of the differences in results between using different data sets can vary, depending on the application and even location. Response: Thank you. We agree and have completely revised the respective section. The important sentence regarding this comment now reads as follows:

“We apply a case study-based approach, in which we provide three illustrative examples of how data with different kinds and levels of uncertainty affect environmental assessments and, by that, the decision-support provided by environmental planning.” (Lines 20-22)

8. Use a subscript for the 2 in CO2 throughout. Response: This has been changed throughout the manuscript. 

9. L92 – I can’t understand this sentence – perhaps rephrase? Response: Thank you for pointing this out; this is now case study number three. The phrasing is now as follows: has been reworded:

“Case study number three, ‘National renewable energy planning’, concerns an environmental planning effort aiming to identify areas for the siting of renewable energies, which should also guide decision-making at the regional level.“(Lines 110-112)

10. L115 – “no significant missing data” – does this mean there is some missing data? If so, please explain. Response: This part of the manuscript has been completely revised. The data this referred to is no longer a part of the analysis that the manuscript is based on. 

11. Figure 1. Please add (a) scale bar (in m, or km, as may be appropriate) and (b) a legend. This figure has been removed in the process of changing the focus of the manuscript.

12. I think it would be useful to point out to the reader that the issue is not just the pixel size of raster data, but the minimum mapping unit area – it is convenient that the CORINE data specifies this so clearly. Response: Thank you for this comment. Although the focus is still on the inherent uncertainties and implications of using different data it is no longer the analysis of the uncertaitnty itself that is the focal point of the manuscript. 

13. L189, 190 and 195/196. The details of the grid cell size should be in Table 1. (I suggest using “grid cell size” or “grid cell dimension”, or even “pixel size”, not “grid width”) Response: We agree with you and have adjusted the table 1 and the text accordingly.For this we have used "grid cell size", thank you.

14. L195 – In my experience, even 50 m is still very coarse for this type of modeling. I think this reinforces the idea that often in modeling we have to use data that don’t entirely fit our purpose. Response: Digital elevation models with a grid cell size of 10m and 25m can also be requested from the Federal Agency for Cartography and Geodesy. We have decided in this investigation for the DEM50 due to computing capacities and agree with you completely that in modeling also data are integrated which are faster to process or whose provision is realized faster. We added a part to the discussion highlighting the need to carefully select appropriate data sets for environmental anaylsis. This part reads as follows:

“Taken together, the results re-emphasize the importance of carefully selecting datasets for landscape and environmental analysis for planning, as the first impression of sufficient data certainty might be misleading, and error-propagation through data application in models can result in substantially misleading results.” (Lines 322-325)

15. L198 – “user’s accuracy”, not “user accuracy” (but seem my comment above about not using the term accuracy). Response: This part of the manuscript has been completely revised.

16. L199 – what is “combined accuracy” – do you just mean just “the range in the user’s accuracy”? – but that doesn’t make sense since the lowest value seems to be 0%, not 82%? Do you perhaps mean the “overall accuracy” (which is defined as the sum of the correctly classified areas in each class, divided by the total area of the map)? Response: This part of the manuscript has been completely revised.

17. Figure 2. The y-axis should be “Agreement”, not “Spatial accuracy” (the latter would be a measure of the spatial registration of the two data sets, which I don’t believe is what you mean). Is it possible to use color – I struggled to tell the different grays apart. Response: This figure is no longer included in the manuscript.

18. L219 – “correlation” – do you perhaps mean “confusion”? Response: This part of the manuscript has been completely revised.

19. Figure 3. Are the legends correct? For example, for Figure 3 (a), my interpretation from the legend (“misclassified GRASSLAND”) is that these are areas that are in reality GRASSLAND but have been misclassified. But I think you actually might mean “Proportion of pixels misclassified as GRASSLAND”? Response: Thank you for this remark, your first interpretation was correct, but this figure is no longer part of the manuscript.

20. L260 “rundue” should be two words. Response: This has been fixed, thank you.

21. L274 – replace “geometric” with “spatial resolution”. Response: This part of the manuscript has been completely revised.

22. L294/295 – I don’t understand the sentence (or the subsequent discussion) and my best interpretation doesn’t seem to be correct. The confusion matrix is a spatial overlay and produces a summary of agreement. It is true however that it does not consider the contiguity of the distribution of pixels in any one class, but I don’t see why that should matter. Response: This part of the manuscript has been completely revised.

23. L300 – yes, it has uncertainty. Since “to a certain degree” are not defined, they can be removed. Response: This part of the manuscript has been completely revised.

24. L302 - The term “geodata uncertainty marker” is not defined and it is not clear any case that one can generalize for other applications – your work shows that even for different areas using the same dataset, the agreement varies. I suggest dropping this. Response: Thank you for this comment. In the process of redesigning the manuscript, this part has been changed and is no longer a focal point of the paper.

25. L321 – I don’t understand why you are saying “the uncertainty is too high” when the “effect is small”? Response: This part of the manuscript has been completely revised.

26. L332 I don’t understand why small land holders would be affected more than large ones. Wouldn’t small ones be too small to show up in the analysis, and therefore be ignored? Perhaps you could elaborate. Response: This part of the manuscript has been completely revised.

27. L345. I suggest not using a term like “better” (which is not defined), but instead simply state the properties that you mean by “better” – e.g. data with a spatial resolution that matches the modeling requirements, or whatever you mean. Response: This part of the manuscript has been completely revised.

28. L359. I don’t think you have analyzed the uncertainty in the products you have generated, as there are many other sources of uncertainty, including in the model itself, not to mention uncertainty in the “ground truth” data. I think instead you have shown that different data sets can produce different results when applied to a model, and therefore it is important to consider what spatial scale (as well as other data characteristics, such as classes mapped, etc) the model assumes for input data. Response: This part of the manuscript has been completely revised.

29. Figure 4. Could you use color here, too, to make it easier to interpret? Response: This figure is no longer included in the manuscript

30. Supporting data. What are the units of measurement in the supporting confusion matrix data tables? Ha, pixels, etc? (if pixels, what size?). The tables are very hard to read because of the decimals (it is hard to get a visual idea of where the confusions lie) and the size of the numbers. Units like square km would make the table a lot more visually understandable, and perhaps make your overall points easier to make. Response: After the revision of the manuscript this supporting data has been removed as it is no longer relevant.

31. Supporting data. Very important – all the user’s and producer’s accuracies don’t seem to be %, though they are labeled as such. I think they are just straight proportions (on a 0-1 scale). Response: After the revision of the manuscript this supporting data has been removed as it is no longer relevant.

32. Supporting data – what does “balance" mean? What is "sum/balance"? Response: After the revision of the manuscript this supporting data has been removed as it is no longer relevant.

---

## [Decision Letter · Decision Letter 1]

31 Aug 2021

PONE-D-21-14058R1

Uncertainties in land use data may have substantial effects on environmental planning recommendations: a plea for careful consideration

PLOS ONE

Dear Dr. Neuendorf,

Thank you for submitting your manuscript to PLOS ONE. After careful consideration, we feel that it has merit but does not fully meet PLOS ONE’s publication criteria as it currently stands. Therefore, we invite you to submit a revised version of the manuscript that addresses the points raised during the review process.

The revisions required are minor. Please, make the necessary changes or provide the required clarifications. If a particular recommendation is not feasible, please provide cogent explanations as to why.

We look forward to receiving your revised manuscript.

Kind regards,

Gerald Forkuor

Academic Editor

PLOS ONE

Journal Requirements:

Reviewers' comments:

Reviewer's Responses to Questions

**Comments to the Author**

1. If the authors have adequately addressed your comments raised in a previous round of review and you feel that this manuscript is now acceptable for publication, you may indicate that here to bypass the “Comments to the Author” section, enter your conflict of interest statement in the “Confidential to Editor” section, and submit your "Accept" recommendation.

Reviewer #1: All comments have been addressed

Reviewer #2: (No Response)

2. Is the manuscript technically sound, and do the data support the conclusions?

Reviewer #1: Yes

Reviewer #2: Yes

3. Has the statistical analysis been performed appropriately and rigorously? 

Reviewer #1: Yes

Reviewer #2: Yes

4. Have the authors made all data underlying the findings in their manuscript fully available?

Reviewer #1: Yes

Reviewer #2: No

5. Is the manuscript presented in an intelligible fashion and written in standard English?

Reviewer #1: Yes

Reviewer #2: No

6. Review Comments to the Author

Reviewer #1: The revised version of the manuscript is greatly improved in terms of clarity and provides an important message for the planning community.

Reviewer #2: The authors have made a good-faith effort to take into account my concerns. My suggestions below are mainly minor and mainly merely a matter of clarifying wording or terms that I find confusing, as unfortunately the revised paper is often a bit confusing.

1. In a number of locations “topographical” is used where it appears “land cover” would be a better term (topographical means the form of the land – e.g. hill, valley, etc). See L57, L137

2. L81-85 – The meaning of this is not clear – can you rephrase?

3. L107 – Adressees – not sure what this means – do you mean “Potential users”?

4. L135 – Actuality – do you mean “currency” (as in the data is up to date)? Or something else?

5. L145 – Pictures – replace with “images”

6. Methods – do you explain how you handle your spatial overlay operations evaluating “consistency” where the pixel size differ? Specifically, how do you resize the pixels?

7. L191 onwards – in the results you refer to “After Walter” and “After Thiele” – I suggest making it clear by including those names in the text here and also explaining explicitly how your methods relate to those sources – e.g. are you following their procedures or using their output or what?

8. L215, Figure 1 and Table 1. The word “consistent” alone does not explicitly explain the meaning that I think you are applying to this term. If I understand it correctly, you do two comparisons: an aspatial comparison of total areas mapped as a certain class, and a spatial overlay operation where you look for agreement in class labels of pixels. You appear to use “consistent” to mean the latter only. I think to make this usage clearer it would help a lot to say “consistent labels” or “consistent labeling” or “consistency in labeling of pixels” or “spatial agreement”. I also think you should state in the methods that you have these two ways of evaluating agreement.

9. L221 and also figure 2. In some cases you use the term “area” where I think you mean “class” or “label”. For example, in L221 – I think the point you are making is that errors in the delineation of certain classes can have a large effect on the model outcome.

10. L227 – “this case” – do you mean “this class”? I’m also confused about “deficits” and “untapped” – it would be clearer to simply talk about “disagreements” and “differences” or “not mapped”.

11. L231 – perhaps rephrase along the lines of “In contrast, the use of ATKIS data results in an underestimation of about 5.8 million tons of potential C02 emissions compared to the reference data.”

12. In the next sentence replace with something like “This is because a total of 3,025 fewer ha were mapped as Level 1…”

13. Table 2. This is a tricky table to read and absorb. I think it would help a lot to eliminate the use of decimals for the t and ha values. This level of precision (1/100 of 1 ha!) is of little purpose when the numbers differ as much as they do (thousands to millions of ha). Important: the “Potential CO2 emission or retention” cell label for the reference Biotope needs units (presumably t).

14. L248 – add a caveat “if we assume the Biotope mapping is correct.”

15. L251 – It wouldn’t be potentially disregarded, but would be disregarded – i.e. rephrase something like “This would lead to a total of 255 ha of areas important to conservation not recognized in the planning recommendations due to the lower detail of the data used.”

16. Table 3 – indicate in the table that Biotope is the reference mapping

17. Table 3 – need to clarify in the headings that it is “class label” that matches, or that the “class is not mapped” – rather than the area that is missed or consistent (which would mean something else).

18. L302 – thematic attributes - do you mean fewer classes?

19. L307- thematic information is lacking – do you mean that if the thematic classes do not match the classes of interest?

20. L312 – 2980 km2 – the significance of the area is hard to understand – I recommend also specifying this as a percentage.

21. L323- easy data – do you mean “easily obtained data??

22. L344 – albeit this would not be necessary – do you mean “that are not actually helpful based on the real land use of the area.”

23. L347 – I don’t understand what this fourth constellation is – do you mean something like “when the potential effort or response required to address the issues raised by the model outcome is large but the data uncertainty is also large?

24. L375-6 – I can’t understand this sentence.

25. L394 – I don’t understand the sentence (“in how far”?) and what does “behavioral mechanisms” mean? Can you rephrase this?

7. PLOS authors have the option to publish the peer review history of their article (what does this mean?). If published, this will include your full peer review and any attached files.

Reviewer #1: **Yes: **Ana Stritih

Reviewer #2: No

---

## [Author Response · Author response to Decision Letter 1]

27 Oct 2021

E: The revisions required are minor. Please, make the necessary changes or provide the required clarifications. If a particular recommendation is not feasible, please provide cogent explanations as to why.

- Thank you very much for the advice. We have done as suggested

R1: The revised version of the manuscript is greatly improved in terms of clarity and provides an important message for the planning community.

- Thank you for the kind words.

R2: The authors have made a good-faith effort to take into account my concerns. My suggestions below are mainly minor and mainly merely a matter of clarifying wording or terms that I find confusing, as unfortunately the revised paper is often a bit confusing.

- Thank you for the kind words and the detailed re-read of the manuscript. 

R2: 1. In a number of locations “topographical” is used where it appears “land cover” would be a better term (topographical means the form of the land – e.g. hill, valley, etc). See L57, L137

- Thank you for this important remark. We changed the wording to “spatial and statistical data” in the first case and to “land use / land cover” data in the second case as we think these are the best fitting options.

R2: 2. L81-85 – The meaning of this is not clear – can you rephrase?

- Thank you for flagging this. We have reformulated this section as follows:

“Finally, environmental planners often presume that an intuitively felt uncertainty may be present in their basic data, but that this will not relevantly affect the response measures. The logic behind this is that planners often strive for multifunctional measures which intentionally target a larger area than monofunctional measures that usually address only smaller areas, for example to safeguard a certain species. Along this line, planners generally presume that the inaccuracies of land use data will not affect the validity of recommended measures which address multiple purposes in an aggregated spatial unit. Furthermore, in case of precautionary measures, the uncertainties of proposals of future land uses seem to be much more relevant than uncertainties stemming from inaccuracies of cartographic land use data.”

R2: 3. L107 – Adressees – not sure what this means – do you mean “Potential users”?

- We have changed this to ‚potential users‘

R2: 4. L135 – Actuality – do you mean “currency” (as in the data is up to date)? Or something else?

- Thanks. We changed the term to ‚currency‘ as kindly suggested. 

R2: 5. L145 – Pictures – replace with “images”

- Changed as suggested

R2: 6. Methods – do you explain how you handle your spatial overlay operations evaluating “consistency” where the pixel size differ? Specifically, how do you resize the pixels?

- We added a section to explain how the raster data was resized. This reads as follows:

„In order to compare the two Digital Elevation Models (input data), the DEM200 was converted to a grid with a grid cell sizes of 50 m. Since the potential areas of the two model runs were also provided as grids with a grid cell size of 50 m, they could be compared without further transformations.” 

R2: 7. L191 onwards – in the results you refer to “After Walter” and “After Thiele” – I suggest making it clear by including those names in the text here and also explaining explicitly how your methods relate to those sources – e.g. are you following their procedures or using their output or what?

- Thank you for flaggining this. We have rephrased the whole section, beginning with: “The case study ‘National renewable energy planning’ explored impacts of using datasets of different uncertainty levels on the potentially suitable sites for renewable energies, taking into account both human and nature-conservation needs [20]. We used the data and results of two model runs, the first one was conducted by Walter et al. 2018 and the second one by Thiele et al. 2021, for an uncertainty analysis in order to quantify the influence of data resolution on identifying opportunity spaces for generating renewable energy...”

R2: 8. L215, Figure 1 and Table 1. The word “consistent” alone does not explicitly explain the meaning that I think you are applying to this term. If I understand it correctly, you do two comparisons: an aspatial comparison of total areas mapped as a certain class, and a spatial overlay operation where you look for agreement in class labels of pixels. You appear to use “consistent” to mean the latter only. I think to make this usage clearer it would help a lot to say “consistent labels” or “consistent labeling” or “consistency in labeling of pixels” or “spatial agreement”. I also think you should state in the methods that you have these two ways of evaluating agreement.

- Thank you for this comment. We added an explanatory sentence to the method section to specify what is meant with consistent in this case.It reads as follows: “We compare resulting maps also in terms of spatial consistency, meaning areas where there is a spatial match of the class labels between the different outputs.”

R2: 9. L221 and also figure 2. In some cases you use the term “area” where I think you mean “class” or “label”. For example, in L221 – I think the point you are making is that errors in the delineation of certain classes can have a large effect on the model outcome.

- We understand the confusion and have rewritten the sentence as follows: “… highlight the importance of a good representation of sites of importance for climate protection. “

R2: 10. L227 – “this case” – do you mean “this class”? I’m also confused about “deficits” and “untapped” – it would be clearer to simply talk about “disagreements” and “differences” or “not mapped”.

- Sorry for the confusion, we have revised the sentence for clarity but did not change the exact wording as it would alter the intended message.It now reads: Surprisingly, using coarser CLC 25 data leads to better results, with a smaller area deficit of around 1,472 hectares that accounts for around 2.8 million t CO2 retention potential untapped.

R2: 11. L231 – perhaps rephrase along the lines of “In contrast, the use of ATKIS data results in an underestimation of about 5.8 million tons of potential C02 emissions compared to the reference data.”

- Thank you, we have rephrased the section

R2: 12. In the next sentence replace with something like “This is because a total of 3,025 fewer ha were mapped as Level 1…”

- Thank you, we have rephrased as suggested. 

R2: 13. Table 2. This is a tricky table to read and absorb. I think it would help a lot to eliminate the use of decimals for the t and ha values. This level of precision (1/100 of 1 ha!) is of little purpose when the numbers differ as much as they do (thousands to millions of ha). Important: the “Potential CO2 emission or retention” cell label for the reference Biotope needs units (presumably t).

- Yes, you are absolutely right. We have implemented both changes as suggested. 

R2: 14. L248 – add a caveat “if we assume the Biotope mapping is correct.”

- We have added a more detailed explanation in the data description section where we state the following: “…as it is the most detailed type of land-use data used for environmental planning in Germany and we assume that it depicts the real circumstances best.” 

R2: 15. L251 – It wouldn’t be potentially disregarded, but would be disregarded – i.e. rephrase something like “This would lead to a total of 255 ha of areas important to conservation not recognized in the planning recommendations due to the lower detail of the data used.”

- Yes, we have done as suggested. Thanks so much for the very useful recommendations. 

R2: 16. Table 3 – indicate in the table that Biotope is the reference mapping

- Thanks, we have done accordingly.

R2: 17. Table 3 – need to clarify in the headings that it is “class label” that matches, or that the “class is not mapped” – rather than the area that is missed or consistent (which would mean something else).

- Thank you. We have added a title for column 2 to clarify what is meant. The column title reads”Type of area considered”. We also added some specification in the cells below where needed for further clarification. 

R2: 18. L302 – thematic attributes - do you mean fewer classes?

- Yes, this is what we mean. We have corrected it accordingly. 

R2: 19. L307- thematic information is lacking – do you mean that if the thematic classes do not match the classes of interest?

- Yes. We revised the sentence to be more comprehensible. 

R2: 20. L312 – 2980 km2 – the significance of the area is hard to understand – I recommend also specifying this as a percentage.

- Thanks for the important comment, we have integrated the following: “Thus, the potential area for onshore wind energy in the second model run is 34% smaller than in the first model run.”

R2: 21. L323- easy data – do you mean “easily obtained data??

- We changed to easily obtained data.

R2: 22. L344 – albeit this would not be necessary – do you mean “that are not actually helpful based on the real land use of the area.”

- Yes, this is what we mean. We changed the sentence as suggested. 

R2: 23. L347 – I don’t understand what this fourth constellation is – do you mean something like “when the potential effort or response required to address the issues raised by the model outcome is large but the data uncertainty is also large?

- Yes, this is what we mean. We now write, as suggested: “… the fourth problem constellation is when the potential effort or response required to address the issues raised by the model outcome is large but the data uncertainty is also large. This applies, for example, even if the effect would be harmless, as… “

R2: 24. L375-6 – I can’t understand this sentence.

- Thank you for pointing this out. Indeed, this sentence is distorted. We have revised it and now write: “The realistic dilemma of using easily available data instead of better but hard to obtain data with uncertain added value in terms of relevance for answering the planning question will not vanish in the near future.”

R2: 25. L394 – I don’t understand the sentence (“in how far”?) and what does “behavioral mechanisms” mean? Can you rephrase this?

- Thank you for flagging this. We have revised it as follows: “Finally, in order to enhance the acceptability of uncertainty communication to decision makers, future research should explore how behavioral interventions such as splitting complex, uncertain decisions into smaller decisions with regard to space and time (29) would increase acceptability.”

---

## [Editor Report · Decision Letter 2]

8 Nov 2021

Uncertainties in land use data may have substantial effects on environmental planning recommendations: a plea for careful consideration

PONE-D-21-14058R2

Dear Dr. Neuendorf,

We’re pleased to inform you that your manuscript has been judged scientifically suitable for publication and will be formally accepted for publication once it meets all outstanding technical requirements.

Kind regards,

Gerald Forkuor, Ph.D

Academic Editor

PLOS ONE
---

## [Editor Report · Acceptance letter]

16 Nov 2021

PONE-D-21-14058R2 

Uncertainties in land use data may have substantial effects on environmental planning recommendations: a plea for careful consideration 

Dear Dr. Neuendorf:

I'm pleased to inform you that your manuscript has been deemed suitable for publication in PLOS ONE. Congratulations! Your manuscript is now with our production department. 

Kind regards, 

on behalf of

Dr. Gerald Forkuor 

Academic Editor

PLOS ONE